# TSP with Predictions: Heatmap to Tour with Provable Guarantees

**Marek Eliáš** [* 1]   **Fabrizio Grandoni** [* 2]   **Adam Polak** [* 1]   **Eleonora Vercesi** [* 3 2]

## Abstract

The Traveling Salesperson Problem (TSP) has long served as a benchmark for evaluating the strength of optimization techniques in the classical theory of algorithms. In recent efforts to apply ML to algorithmic problems, TSP has also become a natural testbed for the development of ML-based techniques. A common approach is to train a neural network to output a heatmap estimating the likelihood of each edge to be part of the optimal tour; however, converting such a heatmap into an actual tour remains a non-trivial and often computationally intensive step. In this work, we propose algorithms for transforming heatmaps into tours with theoretical guarantees linking the achieved approximation ratio to the quality of the provided heatmap. In the spirit of *algorithms with predictions*, our results can be described as $(1 + 2\eta/\text{OPT})$-approximation algorithms, where $\eta$ denotes the L1 distance between the prediction (heatmap) and an optimal solution (tour). Since the previous works lack such explicit guarantees, we compare our approach against them experimentally.

## 1. Introduction

We study the Traveling Salesperson Problem (TSP), one of the key problems in combinatorial optimization: We are given a road network between $n$ cities represented by an undirected edge-weighted graph $G$. Our task is to find a closed walk in $G$ of minimum total weight that visits every city (at least once).

Our main result is an algorithm that takes as input a TSP instance and additionally, for each edge, a number between 0 and 1 denoting how likely this edge is to belong to an optimal TSP tour. These additional numbers can be understood as *predictions* or a *heatmap*. The algorithm outputs a tour that is guaranteed to be at most $2\eta$ longer than an optimal tour, where $\eta$ denotes the L1 distance between the predictions vector and a groundtruth optimal tour.

Our contribution can be understood in two contexts: a theoretical context of the so-called algorithms with predictions (a.k.a. learning-augmented algorithms) (Mitzenmacher & Vassilvitskii, 2020; Lindermayr & Megow, 2022), and a practical context of the so-called solution search stage in the neural combinatorial optimization pipeline (Joshi et al., 2022).

**Algorithms with predictions.** This line of work tries to combine provable worst-case guarantees of classical algorithms with great beyond worst-case performance of ML models. The idea is to have a classical algorithm taking as additional input an uncertain hint – *prediction* of an ML model – and have its performance (e.g., running time or solution cost) bounded by a function of the error of the prediction. Despite almost 400 papers (Lindermayr & Megow, 2022) developed over the last few years, approximation algorithms for TSP have not been studied in this paradigm. We address this gap in the literature.

**Neural combinatorial optimization.** It is a big question if and how neural networks (NNs) can help solving combinatorial optimization problems with many hard constraints, which are usually hard to impose on NN's output (Bengio et al., 2021). TSP has long served as a benchmark for combinatorial optimization techniques, and it has a long history of hand-optimized solvers (Applegate & Cook, 2006; Helsgaun, 2000). So it is interesting to understand if NNs can beat them. A common neural optimization approach to TSP – formalized by Joshi et al. (2022) into a 5-stage pipeline – is to have a (graph) NN that assigns to each edge the probability that it belongs to an optimal solution (the so-called heatmap), followed by a "solution search" algorithm that turns these probabilities into an actual solution (a tour). Initially, researchers used very simple solution search algorithms, e.g., greedy search or beam search (Vinyals et al., 2015; Khalil et al., 2017; Joshi et al., 2019; Li et al., 2024), which have no theoretical guarantees linking the NN

---

[*]Equal contribution [1]Bocconi University, Milan, Italy [2]IDSIA, USI-SUPSI, Lugano, Switzerland [3]Università della Svizzera italiana, Lugano, Switzerland. Correspondence to: Marek Eliáš <marek.elias@unibocconi.it>, Fabrizio Grandoni <fabrizio@idsia.ch>, Adam Polak <adam.polak@unibocconi.it>, Eleonora Vercesi <eleonora.vercesi@usi.ch>.

*Proceedings of the $43^{rd}$ International Conference on Machine Learning*, Seoul, South Korea. PMLR 306, 2026. Copyright 2026 by the author(s).

performance to the final solution cost. Our algorithm is a drop-in replacement for the solution search stage, and tends to work better than greedy in practice (see Section 7).

More recently, researchers started using Monte Carlo Tree Search (MCTS) for solution search (Fu et al., 2021), making it the heaviest part of the pipeline, heavier than the NN itself. Actually, Xia et al. (2024) argue that then NNs become useless, because one can as well replace the input to MCTS with a simple heuristic (the softmax of negative edge weight). Overall, MCTS does lead to better results, but this does not answer the original question of how NNs can help in combinatorial optimization.

To summarize, people started using MCTS in the pipeline because greedy search was not good enough, but with MCTS the NN becomes useless. Our algorithm is a better replacement for greedy search, but, unlike MCTS, it heavily uses the NN output while itself remains lightweight. Hence, our algorithm creates an opportunity for better NNs to become useful overall.

## 1.1. Our results

We propose algorithms for TSP that, given *any* vector of probabilities (a.k.a. heatmap) of each edge to belong to an optimal solution, computes a feasible *TSP tour*. This tour is provably *near-optimal* whenever the given heatmap is *sufficiently accurate*. We experimentally compare the solutions produced by our algorithms on Euclidean instances of TSP with existing heatmap-based techniques in the literature, considering heatmaps produced by NNs as well as by standard heuristics.

**Definition of TSP.** Let $G = (V, E)$ be a complete undirected graph (i.e., $E = \binom{V}{2}$) and $w \colon E \to \mathbb{R}_+$ the weights of its edges satisfying the triangle inequality[1] ($w(uv) \leq w(uz) + w(zv)$ for all $u, v, z \in V$). We want to find the optimal TSP tour, i.e., the Hamiltonian circuit on $G$ with the smallest weight.

Consider a heatmap $p \in [0, 1]^E$. Ideally, we want $p$ to be the characteristic vector $\chi^{X^*}$ of some optimal TSP tour $X^*$. We define the prediction error of $p$ with respect to $X^*$ as

$$\eta = \eta(p, X^*) = \sum_{e \in E} \left| p_e - \chi^{X^*}(e) \right| w(e).$$

Our algorithm satisfies the following:

**Theorem 1.1.** *Let $G = (V, E)$ be a complete graph on $n$ vertices with edge weights $w \colon E \to \mathbb{R}_+$ satisfying triangle*

---

[1]Without the triangle inequality, TSP is known to be inapproximable, i.e., no polynomial-time algorithm can achieve a finite approximation ratio for non-metric TSP, unless P=NP (Sahni & Gonzalez, 1976). Unfortunately, this hardness persists even when predictions with arbitrarily small error are available, see Appendix D.4.

*inequality. There is an algorithm running in time $O(n^3)$ which, receiving a heatmap $p \in [0, 1]^E$ with error $\eta$ with respect to some TSP tour $X^*$, finds a TSP tour $X$ of weight*

$$\mathbb{E}[w(X)] \leq w(X^*) + 2\eta.$$

In fact, our algorithm works with a prediction $P \subseteq E$, where each $e \in E$ is included in $P$ independently with probability $p$. Theorem 1.1 can be equivalently stated in terms of $\eta(P, X^*) := \eta(\chi^P, X^*) = w(P \Delta X^*)$, where $\mathbb{E}[\eta(P, X^*)] = \eta(p, X^*)$. In Section D.1, we show that the linear dependence on $\eta$ is necessary.

The statement of Theorem 1.1 holds for all TSP tours $X^*$ simultaneously (optimal or not), which can be interpreted as follows: As far as $p$ has a small error with respect to some TSP tour of a small weight, the algorithm will also find a tour of a small weight. An alternative way of stating this guarantee is

$$\mathbb{E}[w(X)] \leq \min\{w(X^*) + 2\eta(p, X^*) \mid X^* \text{ is a tour in } G\}.$$

Our algorithm is based on the classical 1.5-approximation algorithm by Christofides (1976): It starts by finding a minimum spanning tree $T$ biased towards edges suggested by $p$. Then, it finds a minimum-weight perfect matching $J$ of the odd-degree vertices in $T$. While efficient implementations of algorithms for minimum-weight perfect matching exist (Kolmogorov, 2009), this step remains the major contribution to the running time of our algorithm. We can replace the exact algorithm for matching with a 2-approximation (Goemans & Williamson, 1995) to achieve a near-linear running time while losing a constant factor only in the dependence on $\eta$. However, this requires, counterintuitively, biasing $T$ towards the heaviest edges suggested by $p$ and a more careful analysis. Note that the input size of a weighted complete graph on $n$ vertices is $\Theta(n^2)$. Even faster running times can be achieved on Euclidean and graphical instances which allow succinct representation, see below.

**Theorem 1.2.** *Let $G = (V, E)$ be a complete graph on $n$ vertices with edge weights $w \colon E \to \mathbb{R}_+$ satisfying triangle inequality. There is an algorithm running in time $O(n^2 \log n)$ which, receiving a prediction $p \in [0, 1]^E$ with error $\eta$ with respect to some TSP tour $X^*$, outputs a TSP tour $X$ of weight*

$$\mathbb{E}[w(X)] \leq w(X^*) + 11\eta.$$

**Euclidean TSP.** This is a special case of TSP which admits a succinct representation of the input complete graph $G = (V, E)$. Each vertex $v \in V$ is a point in the Euclidean plane and from such input of size $\Theta(n)$ we can compute $w(uv) = \|u - v\|_2$ for each pair $u, v \in V$. Theorem 1.1 holds in this setting as stated and we can improve the running time of the algorithm in Theorem 1.2

to $O((n+|P|)\operatorname{polylog} n)$ using the fast algorithm for Euclidean MST (Shamos & Hoey, 1975) and an approximation algorithm for Euclidean matching (Varadarajan & Agarwal, 1999).

**Graphical TSP.** Here, $G$ is a connected graph and $w\colon E \to \mathbb{R}^+$ is arbitrary (Cornuéjols et al., 1985). In this case, an optimal tour may need to traverse some edges twice and visit some vertices multiple times. Every instance of metric TSP can be seen as an instance of graphical TSP on the same complete graph, since repetitions of vertices can be eliminated without increasing the weight of the tour. On the other hand, any instance of a graphical TSP on a graph $G$ can be modeled as an instance of metric TSP on the metric closure of $G$ which is a complete graph whose edge weights represent the shortest path metric on $G$. We refer to (Traub & Vygen, 2024) for more details. The most interesting case is when $G$ is sparse and can be represented in the space much smaller than its metric closure. Therefore, we want an algorithm with near-linear running time in the size of $G$ instead of its metric closure.

Since tours in this setting may pass some edges several times, we assume that the predictor provides us information about the multiplicities. The algorithm receives a multiset $P$ of edges which are likely to be used by an (optimal) tour $X^*$ (that is also a multiset) which may be derived, e.g., from the output of some NN. Information about the multiplicities of the edges is necessary to achieve near-linear running time, see Section 6 for details. The error $\eta$ of prediction $P$ with respect to $X^*$ is defined as

$$\eta := w(P \Delta X^*) = \sum_{e \in E} |m_P(e) - m_{X^*}(e)|\, w(e), \quad (1)$$

where $m_S(e)$ denotes the multiplicity of $e$ in the multiset $S$.

**Theorem 1.3.** *Let $G = (V, E)$ be a graph on $n$ vertices and $m$ edges with weights $w\colon E \to \mathbb{R}_+$. There is an algorithm running in time $O(m \log n)$ which, receiving a prediction $P$ with error $\eta$ with respect to some TSP tour $X^*$, outputs a TSP tour $X$ of weight*

$$w(X) \le w(X^*) + 4\eta.$$

Interestingly, if we do not require near-linear running time, predicting edge multiplicities is not necessary. Algorithm from Theorem 1.1 can be extended to this setting, preserving both its running time and approximation guarantees.

## 2. Related Work

**Algorithms with predictions.** Also known as learning-augmented algorithms (see the survey by Mitzenmacher & Vassilvitskii (2020) and updated list of papers by Linder-mayr & Megow (2022)), have been first studied for *online* problems (Lykouris & Vassilvitskii, 2021; Purohit et al.,

2018), with the premise that ML models may predict future data. Starting with the seminal work by Dinitz et al. (2021), predictions are also used to improve offline (static) algorithms, which becomes useful when solving multiple similar instances and an ML model can discover a common structure. Among those, there are only few works that concern approximation algorithms. The closest to our work is the one by Antoniadis et al. (2025) about subset selection problems. Unfortunately, their algorithm does not preserve triangle inequality of the input instance and therefore it does not apply to TSP (see also Remark 4.1). Bampis et al. (2025) proposed an algorithm for permutation problems including TSP running in (large) polynomial time which finds an optimal solution in the setting with $\epsilon$-accurate predictions where errors occur independently at random.

**Neural combinatorial optimization.** In this vast area of research (see, e.g., Bengio et al. (2021) and Joshi & Anand (2022)) TSP plays a central role. Papers most closely related to our work are discussed in Sections 1 and 7.

Beyond the heatmap-to-tour paradigm, several alternative approaches have emerged. Reinforcement learning (RL) has been explored extensively, with methods like POMO (Kwon et al., 2020) and other policy-based approaches (Kim et al., 2021) showing promising results. Transformer-based architectures have also gained attention, including MatNet (Kwon et al., 2021) and BQ-NCO (Drakulic et al., 2023). Additional architectural innovations are explored in Sym-NCO (Kim et al., 2022) and DIMES (Qiu et al., 2022).

More recently, *Preference Optimization* (a technique that transforms quantitative reward signals into qualitative preference comparisons) has gained attention as an alternative to RL-based approaches (Pan et al., 2025a; Fang et al., 2026; Liao et al., 2025). Concurrently, other works have explored hybrid architectures that blend global predictions with local solution construction (Ye et al., 2024; Ma et al., 2025a), while further methods divide the search space into sub-problems solved after by neural solvers in sequence (Zhou et al., 2025).

Another emerging paradigm aims to train general-purpose neural solvers capable of addressing diverse combinatorial problems with a single model (Drakulic et al., 2025; Pan et al., 2025b). For a comprehensive analysis of these various architectural choices and their roles in neural combinatorial optimization, particularly for TSP, we refer the reader to (Li et al., 2025).

**TSP.** The problem is known to be NP-hard, and even hard to approximate if distances do not satisfy the triangle inequality and nodes cannot be visited twice (Karp, 1972). In the metric case, there are simple 2-approximation algorithms (Rosenkrantz et al., 1977); (Christofides, 1976) and

(Serdyukov, 1978) proposed a 1.5-approximation algorithm which was only improved recently to $1.5 - 10^{-36}$ (Karlin et al., 2021). We refer to the textbooks by Applegate et al. (2006) and Traub & Vygen (2024).

**Fast algorithms for Euclidean graphs.** Euclidean setting often allows algorithms with improved performance. In particular, Arora (1998) proposed an approximation scheme for Euclidean TSP, Euclidean minimum spanning tree can be computed in time $O(n \log n)$ using Delaunay triangulations (Shamos & Hoey, 1975), and Varadarajan & Agarwal (1999) proposed an $(1 + \epsilon)$-algorithm for Euclidean min-weight perfect matching running in time $O(\epsilon^{-3} n \log^6 n)$.

## 3. Notation and Preliminaries

In some parts of the paper, we work with multisets, extending classical set operations as follows. Let $F$ be a multiset from a universe $U$, and $m_F(e) \in \mathbb{N}$ denote the multiplicity of $e \in U$ in $F$. Then the following holds for any 3 multisets $S$, $A$, and $B$ of $U$ and any $e \in U$. If $S = A \cap B$, then $m_S(e) = \min\{m_A(e), m_B(e)\}$. If $S = A + B$, then $m_S(e) = m_A(e) + m_B(e)$. If $S = A - B$, then $m_S(e) = \max\{m_A(e) - m_B(e), 0\}$. We also define the symmetric difference $A \Delta B = (A - B) + (B - A)$.

Given an input graph $G = (V, E)$, we denote by $n = |V|$ the number of its vertices and by $m = |E|$ the number of its edges. We say that edge weights $w \colon E \to \mathbb{R}^+$ are metric if they satisfy triangle inequality, i.e., if we have $w(uv) \leq w(uz) + w(zv)$ for each $u, v, z \in V$. For a set $F \subseteq E$, we denote by $\mathrm{odd}(F)$ the set of vertices in $(V, F)$ with an odd degree. We say that $G = (V, E)$ is Eulerian, if $\mathrm{odd}(E) = \emptyset$. Given a set $S \subseteq V$ of even size, we call $J \subseteq E$ an $S$-join if $\mathrm{odd}(J) = S$. We call $J \subseteq E$ a perfect matching of $S$ if, in $(V, J)$, the degree of each $v \in S$ is equal to 1 and the degree of each $v \in V \setminus S$ is equal to 0. Note that a perfect matching of $S$ is a special case of an $S$-join. In graphs with metric weights, we can assume that all $S$-joins of minimum weight are perfect matchings.

A closed walk in $G$ passing through each vertex at least once is called a *tour*, let $X$ denote the multiset containing each edge $e \in E$ as many times as it is traversed by the tour. It is well-known and easy to prove that $(V, X)$ is Eulerian. If $G$ is a complete graph with metric edge weights and $F$ is a multiset of edges such that $(V, F)$ is Eulerian, then there is a Hamiltonian cycle $C \subseteq F$ of weight $w(C) \leq w(F)$ which can be found using the standard short-cutting procedure, see e.g. (Traub & Vygen, 2024, Lemma 1.7).

## 4. Algorithms for Metric TSP

Our algorithms receive in input a *prediction* $P \subseteq E$, which is ideally the set of edges of an optimal tour $X^*$. Notice that

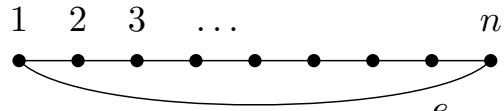

*Figure 1.* Example instance of a TSP where the vertices are points on a line. Only the edges of the optimal tour $X^*$ are drawn. In the case with perfect prediction, i.e., $P = X^*$, Algorithm 1 may choose the path from 1 to $n$ as $T$ (there are multiple possibilities), $S = \{1, n\}$ and $J = \{e\}$ contains the edge $e$ connecting $n$ and 1. Then $T + J = X^*$ and no short-cutting is needed.

in general we allow $P$ to be any set of edges, in particular $P$ might induce a disconnected graph, a graph with multiple cycles etc. In the settings where we are given a heatmap $p \in [0, 1]^E$ in input, we simply sample each edge $e \in E$ independently with probability $p_e$, and let $P$ be the set of sampled edges. We remark that $\mathbb{E}[\eta(P, X^*)] = \eta(p, X^*)$, therefore it is sufficient to upper bound the cost of the produced solution in terms of $\eta = \eta(P, X^*)$.

### 4.1. Algorithm 1 and proof of Theorem 1.1

We describe our algorithm proving Theorem 1.1. Our algorithm is based on the classical 1.5-approximation algorithm by (Christofides, 1976). First, it constructs a minimum spanning tree (MST) $T$ on $G$ which is biased towards edges in the prediction $P$ in the sense that each edge in $P$ is treated as if its weight was 0. This can be easily done in near-linear time. Having the spanning tree $T$, it computes the minimum-weight $\mathrm{odd}(T)$-join $J$ which, in the metric case, is the minimum weight perfect matching of $\mathrm{odd}(T)$. We compute $J$ using the classical algorithm (Gabow, 1973; Edmonds & Johnson, 1973) with running time[2] $O(n^3)$ treating predicted and not predicted edges equally, i.e., we always count with their original weights $w$.

The multiset $T + J$ where edges contained in both $T$ and $J$ are taken twice is connected (because $T$ is a spanning tree) and Eulerian (each $v \in V$ is contained in an even number of edges in $T + J$), since $J$ is an $\mathrm{odd}(T)$-join, i.e., $\mathrm{odd}(J) = \mathrm{odd}(T)$. Using the classical short-cutting procedure, see e.g. (Traub & Vygen, 2024, Lemam 1.7), we can construct a Hamiltonian circuit of weight no greater than the weight of $T + J$ in time $O(|T + J|)$. See Algorithm 1 for the summary.

*Remark* 4.1. We have to disregard the prediction $P$ when computing $J$. If we have computed $J$ w.r.t. the modified weights $w'$, i.e., each $e \in P$ having weight 0 in the style of Antoniadis et al. (2025), the approximation ratio of our algorithm would be no better than 2 even with prediction error arbitrarily small, see Section D.3.2.

*Remark* 4.2. There is a simple 2-approximation algorithm

---

[2]One can also solve the problem in time $O(n^{2.5} \log W)$ where $W$ is the largest integer weight (Duan et al., 2018).

---

**Algorithm 1:** Algorithm for metric TSP

---

**Input:** $G = (V, E)$ s.t. $E = \binom{V}{2}$, $w \colon E \to \mathbb{R}_+$, $P \subseteq E$
**Output:** TSP tour over $G$

**for** $e \in E$ **do**
   | $w'(e) := 0$ if $e \in P$
   | $w'(e) := w(e)$ otherwise.

$T :=$ MST on $G$ w.r.t. the modified weights $w'$
$S :=$ set of odd vertices in $T$
$J :=$ min-weight $S$-join w.r.t. the original weights $w$
Output: short-cutting of the Eulerian multiset $T + J$

---

called double-tree (Rosenkrantz et al., 1977), which does not require the costly computation of an $\mathrm{odd}(T)$-join $J$. Instead, it outputs the short-cutting of $T+T$ which is clearly Eulerian. Unfortunately, if our algorithm used $T+T$ instead of $T + J$, its approximation ratio would be no better than 2 even with $\eta$ approaching 0, see Section D.3.1.

Theorem 1.1 is implied by the following statement and the fact that each step of Algorithm 1 can be performed in near-linear time except for the construction of $J$ which requires time $O(n^3)$.

**Theorem 4.3.** *Let $G = (V, E)$ be a complete graph on $n$ vertices with edge weights $w \colon E \to \mathbb{R}_+$ satisfying triangle inequality. Let $X^*$ be a TSP tour on $G$ and $P \subseteq E$ a prediction with error $\eta$ with respect to $X^*$. Then the cost of the tour $X$ produced by Algorithm 1 on $G$ with prediction $P$ is at most $w(X^*) + 2\eta$.*

Figure 1 describes the steps of Algorithm 1 on a simple example with a perfect prediction, showing that ideally we have $T \subseteq X^*$, i.e., $T = X^* \cap T$, and $J = X^* \setminus T$. In our analysis, we show that prediction errors can make both $T$ and $J$ deviate from $X^* \cap T$ and $X^* \setminus T$ respectively, but the increase of their weight is proportional to the weight of the mispredicted edges as formulated in the lemmas below. We use the following notation. Given a prediction $P \subseteq E$ and a tour $X^*$, we denote $H^+ = P \setminus X^*$ the set of false positives and $H^- = X^* \setminus P$ the set of false negatives. We have $\eta = w(H^-) + w(H^+)$ and $P = (X^* \setminus H^-) \cup H^+$.

**Lemma 4.4.** *One has:*

$$w(T \setminus P) \le w(H^-) \text{ and} \tag{2}$$
$$w(T) \le w(X^* \cap T) + w(H^+) + w(H^-). \tag{3}$$

*Proof.* Note that $P \cup H^- \supseteq X^*$ is a connected graph and $w'(e) = 0$ for any $e \in P$. Since $T$ is an MST with respect to $w'$, we have

$$w(T \setminus P) = w'(T) \le w'(P \cup H^-) = w(H^-).$$

This already implies (2). To prove (3), we write

$$
\begin{aligned}
w(T) &= w(T \cap P) + w(T \setminus P) \\
&\le w(X^* \cap T) + w(H^+) + w(H^-),
\end{aligned}
$$

where the inequality follows from $T \cap P \subseteq (T \cap X^*) \cup (T \cap H^+) \subseteq (T \cap X^*) \cup H^+$ and (2). $\square$

**Lemma 4.5.** *One has:*

$$w(J) \le w(X^* \setminus T) + w(H^+) + w(H^-).$$

*Proof.* We show that, for $S = \mathrm{odd}(T)$, there is an $S$-join, namely $J' := X^* \Delta T$, that satisfies the desired bound. First, we prove that $J'$ is an $S$-join, i.e., equivalently that $T + J'$ has all degrees even. We can write

$$
\begin{aligned}
T + J' &= \big((T \cap X^*) \cup (T \setminus X^*)\big) + \big((X^* \setminus T) \cup (T \setminus X^*)\big) \\
&= X^* + \big((T \setminus X^*) + (T \setminus X^*)\big),
\end{aligned}
$$

because $(T \cap X^*) \cup (X^* \setminus T) = X^*$. In other words, $T + J'$ is a sum of the tour $X^*$ (Eulerian) and $(T \setminus X^*) + (T \setminus X^*)$ (each edge taken twice). Therefore, $\mathrm{odd}(J') = \mathrm{odd}(T) = S$ and $J'$ is an $S$-join.

Now, it is enough to bound the weight of $J'$. We have $w(J') = w(X^* \setminus T) + w(T \setminus X^*)$, where

$$
\begin{aligned}
w(T \setminus X^*) &= w((T \setminus X^*) \cap P) + w((T \setminus X^*) \setminus P) \\
&\le w(H^+) + w(T \setminus P) \overset{(2)}{\le} w(H^+) + w(H^-).
\end{aligned}
$$

The first inequality follows from $(T \setminus X^*) \cap P \subseteq P \setminus X^* = H^+$ and from $(T \setminus X^*) \setminus P \subseteq T \setminus P$. Thus

$$w(J) \le w(J') = w(X^* \setminus T) + w(H^+) + w(H^-). \quad \square$$

*Proof of Theorem 4.3.* Lemmas 4.4 and 4.5 imply

$$w(T) + w(J) \le w(X^*) + 2w(H^+) + 2w(H^-).$$

Short-cutting of $T + J$ can only decrease the weight of the resulting tour. $\square$

### 4.2. Near-linear time algorithm for metric TSP

Finding the min-weight perfect matching $J$ is the most computationally intensive step in Algorithm 1. An obvious way to speed-up the algorithm is to replace this step with a 2-approximation algorithm by Goemans & Williamson (1995) which runs in near-linear time. Unfortunately, this does not lead to approximation ratio smaller than 1.5 even with very small $\eta$. An example similar to Remark 4.2 can be found in Section D.3.3. Here, we focus on the points in the analysis of the previous section which would fail, in order to motivate the steps performed by our next algorithm.

Consider the example in Figure 1. We have $n$ points on the line and a perfect prediction $P$ of the optimal tour $X^*$ and a spanning tree $T$ that omits the heaviest edge $e \in X^*$ such that $w(e) = w(X^*)/2$. Then, losing a factor of 2 in the bound of Lemma 4.5 due to the approximation ratio of (Goemans & Williamson, 1995), we would have $w(T) + w(J) \le w(X^* \cap T) + 2w(X^* \setminus T) = \frac{3}{2}w(X^*)$, i.e., an 1.5-approximation even with perfect predictions.

---

**Algorithm 2:** Near-linear time algorithm for metric TSP

---

**Input:** $G = (V, E)$ s.t. $E = \binom{V}{2}$, $w \colon E \to \mathbb{R}_+$, $P \subseteq E$
**Output:** TSP tour over $G$

**for** $e \in E$ **do**
  $w'(e) := -w(e)$ if $e \in P$
  $w'(e) := w(e)$ otherwise.
$T :=$ MST of $G$ w.r.t. the modified weights $w'$
$S :=$ set of odd vertices in $T$
**if** $|S| = 2$ **then**
  choose $J := \{uv\}$, where $\{u, v\} = S$
**else**
  $J :=$ 2-apx min-weight perfect matching of $S$ w.r.t. $w$
Output: short-cutting of the Eulerian multiset $T + J$

---

In order to avoid the previous example, we bias $T$ towards the heaviest edges in $P$. In particular, we look for the MST with the signs of the weights of the edges in $P$ flipped. Intuitively, we are trying to make sure that the heaviest edges in $P$ are contained in $T$ and we pay for them only once, while the weight of $X^* \setminus T$ where we suffer the multiplicative factor of 2 will be small.

Unfortunately, $X^* \setminus T$ is always non-empty and contains a part of the optimal solution $X^*$ whose weight we do not want to pay twice. Somewhat surprisingly, we can show that whenever $|S| > 2$ ($\eta > 0$ in such a case), $w(J)$ can be bounded purely in terms of $\eta$, see the lemma below. This means that the approximation ratio of (Goemans & Williamson, 1995) will affect only the constant in front of $\eta$. Fortunately, this is already enough, since finding the exact min-weight perfect matching is trivial with $|S| = 2$ (it is just the edge connecting the two vertices in $S$). Our algorithm is summarized in Algorithm 2. All the steps can be performed in near-linear time.

**Lemma 4.6.** If $|S| > 2$, then $w(J) \le O(\eta)$.

Lemma 4.6 is the crucial part of our analysis. We bound the weight of $J$ using an explicit $\mathrm{odd}(T)$-join $J' = X^* \Delta T$. The proof is most difficult when $\eta$ is relatively small and the most tricky part is in fact accounting for the weight of the lightest edge. We define $e^*$ as the edge in $X^*$ with the smallest weight. First, we ignore $e^*$ and consider $Y^* = X^* \setminus \{e^*\}$.

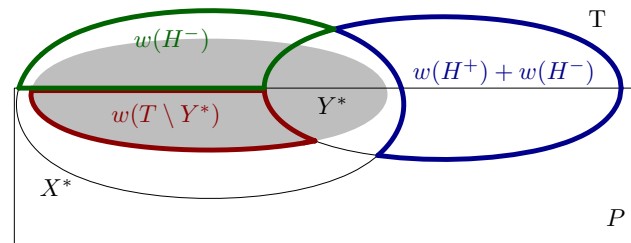

*Figure 2.* Partition of $J' = X^* \Delta T$ together with the bound on each part. In particular, we have $w(T \setminus X^*) \le w(H^+) + w(H^-)$, $w((X^* \setminus T) \setminus P) \le w(H^-)$, and $w((Y^* \setminus T) \cap P) \le w(T \setminus Y^*)$.

We partition $J'$ into parts as in Figure 2, bounding the weight of each part separately, as shown in the figure. The flipped sings of the weights of the edges in $P$ are used in the bound $w((Y^* \setminus T) \cap P) \le w(T \setminus Y^*)$. However, this bound is not yet sufficient to prove Lemma 4.6. We need to bound $w((X^* \setminus T) \cap P)$ instead, and there might be $e^*$ missing on the left- or right-hand side. We need to consider several cases depending on whether $e^*$ belongs to $P$ or not and, if yes, whether it belongs to $T$. The details and the full proof of Theorem 1.2 can be found in Section A.

## 5. Euclidean TSP

In Euclidean TSP, the input is given as coordinates of the $n$ vertices of the complete graph $G$ in the Euclidean plane and the weight of the edge between two vertices $u, v \in V$ is computed as a Euclidean distance between them. In particular, the size of the input is linear in $O(n + |P|) \le O(n)$ for any prediction of reasonable quality. This is much more succinct representation than the vector of edge weights which has $\Theta(n^2)$ coordinates.

We can find $T$ in time $O((n + |P|) \log n)$ as follows: We construct a Delaunay triangulation $D$ of $V$ (has size $O(n)$ and can be found in time $O(n \log n)$), then we find $T$ as an MST of $(V, P \cup D)$. It is well known that an Euclidean MST uses only the edges of $D$ (Shamos & Hoey, 1975). It is not difficult to show that if we modify the weights of edges in the set $P$, the MST will use only edges in $P \cup D$. A simple proof of the following proposition is in Section B.

**Proposition 5.1.** *Let $D \subseteq \binom{V}{2}$ be a Delaunay triangulation of $V$. Then $D \cup P$ contains an MST $T$ with respect to $w'$.*

A $(1+\epsilon)$-approximation of the min-weight perfect Euclidean matching can be found in time $O(\epsilon^{-3} n \log^6 n)$ (Varadarajan & Agarwal, 1999). We find $J$ using their algorithm with $\epsilon = 2$. This way, each step of Algorithm 2 can be implemented in time $O((n + |P|) \operatorname{polylog} n)$ for Euclidean TSP.

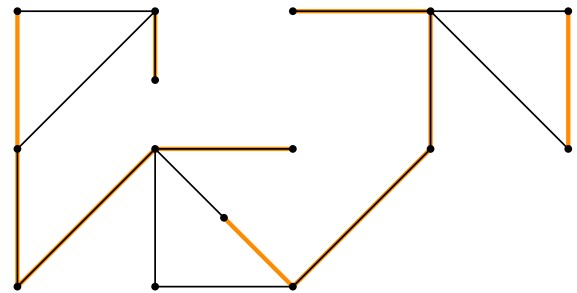

*Figure 3.* Solution of a graphical TSP. Only the edges used by the optimal TSP tour are drawn, you can imagine all the other edges to have a very large weight or be missing, since triangle inequality does not necessary hold in graphical TSP. The tour can be decomposed as $X^* = T + J$, where the edges of the spanning tree $T$ are drawn black, the edges of the T-join $J$ are drawn orange, and the edges drawn both black and orange are traversed twice by the tour.

## 6. Graphical TSP

In graphical TSP, the optimal TSP tour may traverse some edges twice unlike in metric TSP. This means that knowing the set of edges $F$ used by the optimal tour, there is still a non-trivial step needed to identify which edges need to be traversed twice to construct the actual tour. Consider the example in Figure 3: the underlying set $F$ of the optimal tour contains all edges drawn in the picture regardless of their color. In order to transform $F$ into a tour, we need to perform a costly operation: compute the optimal T-join of the odd vertices in $(V, F)$ – edges drawn both black and orange in Figure 3 which will be traversed twice in the resulting tour. Turing $F$ into an optimal solution requires computing an optimal $\mathrm{odd}(F)$-join which is computationally demanding.

In order to achieve near-linear running time, we need the help of the prediction when searching for the T-join. Therefore we require the prediction $P$ to be a multiset containing the information on how many times is each edge traversed and we define the prediction error as in Equation (1).

Our algorithm for graphical TSP is summarized in Algorithm 3. Similarly to Algorithm 1, it finds an MST $T$ biased towards edges in $P$ (the weight of each $e \in P$ is set to 0). Then, it finds a 2-approximate $\mathrm{odd}(T)$-join $J$ biased towards the edges in $P - T$, i.e., the edges in $P$ not used by $T$ or appearing in $P$ more than once have weight 0.

Ideally, we would have a perfect prediction $P = X^* = T + J$ as in Figure 3 and $P - T = X^* - T$ would be used as a prediction for finding $J$. In our analysis, we again consider a reference $\mathrm{odd}(T)$-join $J' = X^* \Delta T$. Our crucial points are that the weight of $P - T$ diverges from the ideal $X^* - T$ by at most $\eta$, while the weight of the unpredicted edges in $(X^* - T) - (P - T)$ is bounded by $\eta$, see Figure 4 for a summary of our bounds. Note that here we consider error $\eta$ w.r.t. to the tour $X^*$ not w.r.t. the optimal T-join

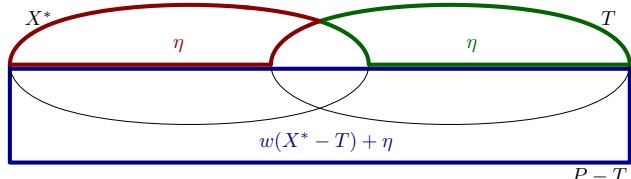

*Figure 4.* Bounding the weight of the reference T-join $J'$. Shows the bounds on the weight of each area highlighted in the picture, namely $w((X^*-T)-(P-T)) \le \eta, w((T-X^*)-(P-T)) \le \eta$, and $w(P - T) \le w(X^* - T) + \eta$.

of $T$. The key observation is that we loose factor of 2 only for the edges in $J' - (P - T)$ whose total weight is $O(\eta)$, while each edge in $P - T$ is taken at most once.

*Remark* 6.1. (Antoniadis et al., 2025) proposed an algorithm with predictions for finding an S-join. We cannot use their analysis since it would provide bounds in terms of the prediction error w.r.t. an optimal $\mathrm{odd}(T)$-join, while we need a bound in terms of the error $\eta$ w.r.t. $X^*$ which may be very different. Consider the following situation: $X^* = T^* + J^*$ and $P = T + J^*$, i.e., the edges in $P - T \subseteq X^*$ do not contribute to $\eta$. However, as a prediction of an $\mathrm{odd}(T)$-join it can be completely wrong, being an $\mathrm{odd}(T^*)$-join of a different tree $T^*$.

---

**Algorithm 3:** Near-linear time algorithm for graphical TSP

**Input:** $G = (V, E)$, $w \colon E \to \mathbb{R}_+$, multiset $P$
**Output:** TSP tour over $G$

**for** $e \in E$ **do**
  | $w'(e) := 0$ if $e \in P$
  | $w'(e) := w(e)$ otherwise.
$T := $ MST on $G$ w.r.t. $w'$
$S := $ set of odd vertices in $T$
**for** $e \in E$ **do**
  | $w''(e) := 0$ if $e \in (P - T)$
  | $w''(e) := w(e)$ otherwise.
$J := $ 2-approximate T-join of $S$ w.r.t. to $w''$
Output: short-cutting of the Eulerian multiset $T + J$

---

Interestingly, Algorithm 1 can be used for graphical TSP as stated, receiving only the prediction $P$ about the ground set of the optimal tour $X^*$. In this case, its running time will be $O(m \log n) + O(|S|^3)$, where the number $|S|$ of the odd vertices in $T$ might be as large as $n$. Details can be found in Section C.2.

## 7. Empirical Results

In this section, we present empirical results that support our theoretical approach. At a high level, we consider probability matrices (*heatmaps*) obtained from different sources

(neural predictors, non-neural predictors, and synthetic constructions) and use them as input to our algorithm. We then evaluate whether this allows us to improve upon standard baselines in terms of optimality gap. The code and data used in the experiments are provided in the Supplementary Material. Here, we provide a high-level overview of our experimental results; full details are deferred to Appendix E.

**The dataset.** We evaluate our model on three different datasets. Dataset $\mathcal{U}$ is the test set of the well-known ML4CO benchmark (Ma et al., 2025b). The dataset consists of instances sampled uniformly from the square $[0,1]^2$ and contains instances of four different sizes: 1280 instances for each of $n \in \{50, 100\}$ and 128 instances for each of $n \in \{500, 1000\}$. Dataset $\mathcal{T}_E$ contains 53 Euclidean 2D TSPLIB (Reinelt, 1991) instances with up to 1300 nodes. Finally, dataset $\mathcal{T}_M$ contains 15 metric *non-Euclidean* instances from TSPLIB. For all instances, we obtain ground-truth optimal tour lengths using Concorde (Applegate & Cook, 2006).

**Data and code availability.** All instances used in our experiments, together with the corresponding predictions, are available on Zenodo.[3] The code for reproducing our results and implementing the main algorithm is available on GitHub.[4]

**Predictions.** We evaluate several types of predictors (Details are provided in Section E.2). Among the **Predictors from NNs** we use the predictions from (Joshi et al., 2022) (**GNN4CO**), which combines a graph neural network encoder with an autoregressive decoder that sequentially builds a tour, (Hudson et al., 2022) (**GNN-GLS**), which predicts for each edge its regret, i.e., the cost of enforcing that edge relative to an optimal solution, and (Sun & Yang, 2023) (**DIFUSCO**), a diffusion-based graph model designed to generate high-quality feasible TSP solutions. We include **SoftDist** (Xia et al., 2024), a non-neural probabilistic baseline where, given a weighted graph, edge probabilities are defined as $p_{ij} = \frac{e^{-w(ij)/\tau}}{\sum_{k \neq i} e^{-w(ik)/\tau}}$, with temperature parameter $\tau$ set with guidance from the original paper. Finally, we evaluate a **Synthetic Predictor (SP)**, constructed by applying perturbation to the optimal solution $X^*$. The noise is governed by the parameter $\varepsilon$.

**Baselines.** Since our approach is largely inspired by Christofides' method, we naturally include Christofides' algorithm (frequently abbreviated as **CHR**) as a baseline. We evaluate the performance of our proposed algorithm against several baseline solution search techniques commonly employed to leverage neural heatmaps (Joshi & Anand, 2022; Hudson et al., 2022; Sun & Yang, 2023). We implement two greedy variants: **G1** builds a Hamiltonian cycle incrementally by starting from an arbitrary node and repeatedly adding the unvisited node with the maximum score $q(e)$; **G2** follows a greedy edge-selection strategy, adding edges in decreasing order of $q(e)$ provided they do not create a subcycle or a node with degree 3, until the tour is complete. We also evaluate a beam search (**BS**), which explores multiple candidate tours in parallel by maintaining the top-$k$ partial paths at each construction step.

**Algorithm 1 vs CHR$^+$.** To derive the edge subset $P$ for Algorithm 1, we evaluate two strategies: sampling $k$ edges (**Alg1**, faithfully representing Algorithm 1), selecting the $k$ edges with the highest probabilities (**Alg1Top**). Furthermore, we test a modified weighting approach, where we set $w'(e) = w(e)(1 - q(e))$, which we denote as CHR$^+$. We set $k = n$, using 30 samples for Alg1 to account for stochasticity. Figure 15 (Appendix E) shows that CHR$^+$ consistently achieves superior performance; thus, we adopt it as our primary decoding strategy. More details can be found in Appendix E.3.

**Our algorithm behaves smoothly as the prediction error decreases.** To control the prediction error, we use SP as the predictor, with $\varepsilon \in [0, 1]$. We first observe that this type of prediction, as expected, performs very poorly with greedy decoding, and hence, comparison is not made in this case (See Appendix E.4). To study the behavior of our algorithm relative to Christofides, we run the full pipeline for $n = 500$ on the entire dataset. Figure 5 demonstrates that our algorithm exhibits smooth behavior as the prediction error decreases, consistently improving upon CHR until the error reaches $\sim \varepsilon = 0.25$. Details on this experiment can be found in Appendix E.4.

**Comparison of different solution search strategies with different predictors.** Table 1 summarizes performance on dataset $\mathcal{U}$ via optimality gaps relative to Christofides. The discrepancy with the original DIFUSCO results (Figure 6) likely stems from inference-time configurations, such as graph sparsification, which were disabled in our setup. Notably, CHR$^+$ is the only strategy that effectively exploits predicted heatmaps to consistently outperform the Christofides baseline. In contrast, G1, G2, and BS fail to yield systematic improvements. These trends remain consistent across TSPLIB instances, both Euclidean and non-Euclidean (Table 2 and Table 3). See Appendix E.5 for further details.

**2-opt improvement.** To further refine the solutions, we apply the 2-opt local search heuristic as a post-processing step, a standard practice for enhancing tour quality (Hudson et al., 2022; Sun & Yang, 2023); All the details are pro-

---

[3] https://zenodo.org/records/20283463
[4] https://github.com/eleonoravercesi/tour_from_neural_predictions

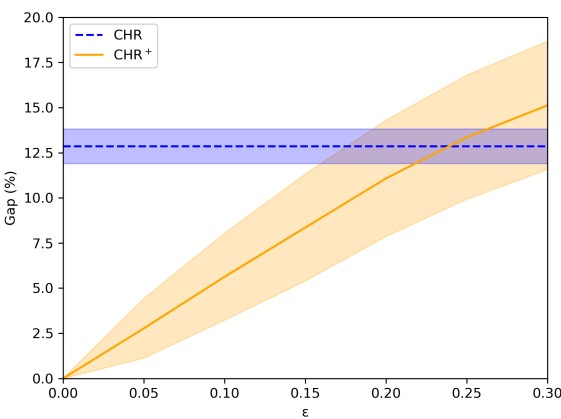

*Figure 5.* Optimality gap (the lower the better) of Christofides and our algorithm (CHR$^+$) on the instances of our dataset with $n = 500$. We report the average percentage gap, with the shaded regions representing $\pm$ standard deviation.

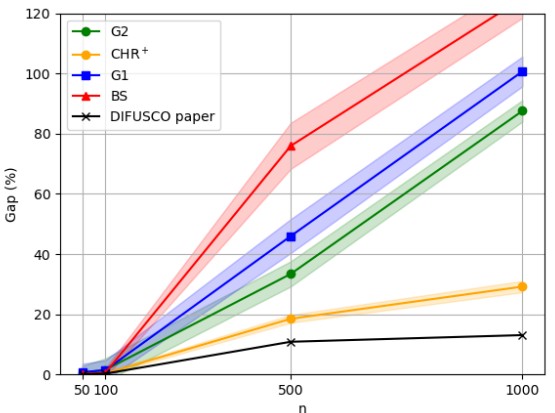

*Figure 6.* Visual representation of the performances of different solution search methods on DIFUSCO: Average percentage gap.

vided in Appendix E.6. Our results demonstrate that CHR$^+$ remains the superior method while introducing negligible computational overhead compared to CHR. This efficiency is evidenced by the average runtimes reported in Table 1 and the comparable number of 2-opt swaps required to reach local optimality, as recorded in Table 4.

## 8. Conclusions

We introduce the *first* learning-augmented approximation algorithm for TSP. Our algorithm converts *any* edge heatmap to a feasible *TSP tour* of weight $\leq$ OPT $+2\eta$, where $\eta$ denotes the L1 distance between the heatmap and some optimal tour of weight OPT. As shown in our empirical results, our approach utilizes heatmaps more effectively than previous works. Moreover, its approximation perfor-

mance improves smoothly with increasing accuracy of the provided heatmap, underscoring the potential for further advancements in high-quality heatmap generation models.

## Acknowledgements

The work of E. Vercesi has been supported by the Swiss National Science Foundation (SNSF) project 200021-212929 / 1 "Computational methods for integrality gaps analysis". The work of F. Grandoni has been partially supported by the SNSF projects 200021-20073 and 200021-236706.

## Impact Statement

This paper presents work whose goal is to advance the field of machine learning. There are many potential societal consequences of our work, none of which we feel must be specifically highlighted here.

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

# A. Analysis of Algorithm 2: Near-Linear Time Algorithm for Metric TSP

We provide full analysis of Algorithm 2 proposed in Section 4.2. Compared to Algorithm 1, Algorithm 2 runs in near-linear time but has slightly worse dependence on prediction error. Instead of the exact algorithm for $\text{odd}(T)$-join, it uses a 2-approximation algorithm by Goemans & Williamson (1995) which runs in time $O(|E|\log n) = O(n^2 \log n)$.

---

**Algorithm 2:** Almost-linear time algorithm for complete graphs (restated for convenience)

---

**Input:** graph $G = (V, E)$, where $E = \binom{V}{2}$, weights $w\colon E \to \mathbb{R}_+$, prediction $P \subseteq E$
**Output:** TSP tour over $G$

**for** $e \in E$ **do** $w'(e) := -w(e)$ if $e \in P$ and $w'(e) := w(e)$ otherwise.
find $T :=$ minimum spanning tree on $G$ w.r.t. modified weights $w'$
$S :=$ set of odd vertices in $T$
**if** $|S| = 2$ **then**
$\quad$ choose $J := \{uv\}$, where $\{u, v\} = S$
**else**
$\quad$ $J :=$ 2-approximate $S$-join w.r.t. to the original weights $w$
output a shortcutting of the Eulerian multiset $T + J$, where edges contained in both $T$ and $J$ are taken twice

---

**Notation:** Let $X^* \subseteq E$ be the edges of an optimal tour of $G$. Since $G$ is a complete graph with metric weights, $X^*$ is a Hamiltonian cycle. Given prediction $P \subseteq E$, we denote $P = (X^* \setminus H^-) \cup H^+$, where $H^-$ and $H^+$ are the sets of false negatives and false positives. We denote by $J' = X^* \Delta T$ the reference $S$-join, with $S = \text{odd}(T)$, which we use to bound the weight of $J$.

**Lemma A.1.** $J' = X^* \Delta T = (X^* \setminus T) \cup (T \setminus X^*)$ *is an $S$-join.*

*Proof.* The claim is equivalent to $T + J'$ having all degrees even. This holds since:

$$T + J' = ((T \cap X^*) \cup (T \setminus X^*)) + ((X^* \setminus T) \cup (T \setminus X^*)) = X^* + ((T \setminus X^*) + (T \setminus X^*)),$$

i.e., $T + J'$ is the union of the tour $X^*$, where all degrees are 2, and $(T \setminus X^*) + (T \setminus X^*)$, where each edge is taken twice. $\square$

**Theorem 1.2 restated.** *Let $G = (V, E)$ be a complete graph on $n$ vertices with edge weights $w\colon E \to \mathbb{R}_+$ satisfying triangle inequality. There is an algorithm running in time $O(n^2 \log n)$ which, receiving a prediction $P \subseteq E$ with error $\eta$ with respect to some TSP tour $X^*$, outputs a TSP tour $X$ of weight*

$$w(X) \le w(X^*) + 7w(H^+) + 11w(H^-).$$

**Lemma A.2.** *One has*

1. $w(T \setminus P) \le w(H^-)$

2. $w(T) \le w(T \cap X^*) + w(H^+) + w(H^-)$.

*Proof.* All the edges $e \in P$ have negative $w'(e)$ while all other edges positive $w'(e)$. Therefore, while there is an edge in $P$ which can still be added to $T$, i.e, some connected component of the graph $(V, P)$ is still disconnected, Kruskal's algorithm will never add an edge $e \notin P$ to $T$. Therefore, $T \cap P$ is a spanning forest of $(V, P)$ and, since $X^* \subseteq P \cup H^-$, we know that $(T \cap P) \cup H^-$ is connected. By the minimality of $T$, we have

$$-w(T \cap P) + w(T \setminus P) = w'(T) \le w'(T \cap P) + w'(H^-) = -w(T \cap P) + w(H^-)$$

which implies $w(T \setminus P) \le w(H^-)$ – the first statement of the lemma.

As for $T \cap P$, we have

$$T \cap P \subseteq (T \cap (P \cap X^*)) \cup (T \cap (P \setminus X^*)) \subseteq (T \cap X^*) \cup (H^+),$$

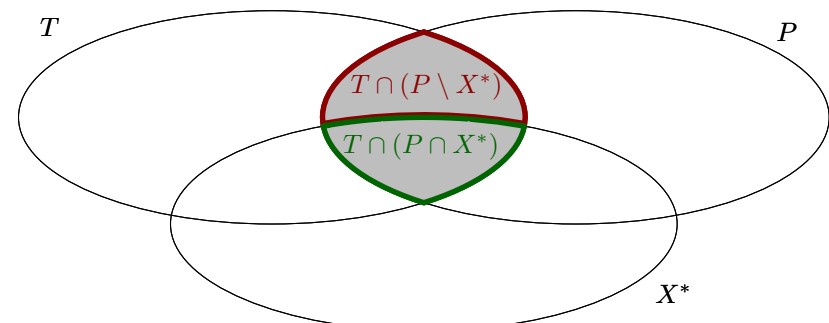

*Figure 7.* Illustration of Lemma A.2

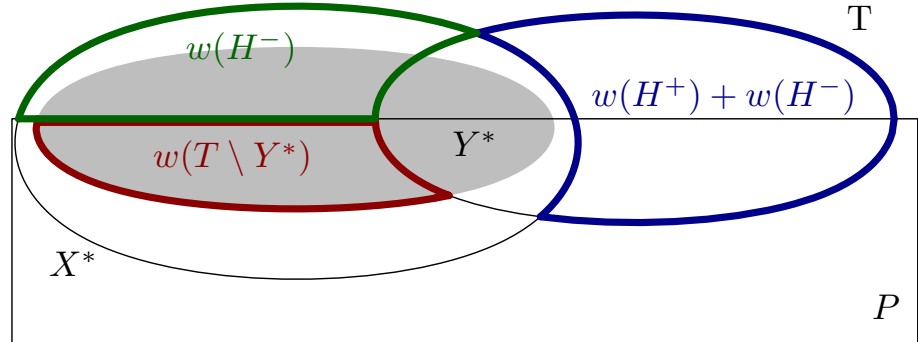

*Figure 8.* Three regions of $J'$ with upper bounds on their weights. The remaining edge $e^*$ is considered separately.

see Figure 7. Therefore, the total weight of $T$ is

$$w(T) = w(T \cap P) + w(T \setminus P) \le w(T \cap X^*) + w(H^+) + w(H^-). \qquad \square$$

The following lemma holds for an arbitrary $S$ but we will need it only for $|S| = 2$.

**Lemma A.3.** *One has*

$$w(J') \le w(X^* \setminus T) + w(H^+) + w(H^-).$$

*Proof.* We have

$$w(J') = w(X^* \setminus T) + w(T \setminus X^*).$$

Since $T \setminus X^* = ((T \setminus X^*) \cap P) \cup ((T \setminus X^*) \setminus P) \subseteq H^+ \cup (T \setminus P)$, we get

$$w(J') \le w(X^* \setminus T) + w(H^+) + w(T \setminus P) \overset{\text{Lem. A.2.(1)}}{\le} w(X^* \setminus T) + w(H^+) + w(H^-). \qquad \square$$

Consider next the case $|S| > 2$. In this case, $J$ is only a 2-approximation of the optimal $S$-join and therefore we can claim only $w(J) \le 2w(J')$. Therefore, to prevent loosing factor of 2 on the weight of the edges in $X^*$, we have to bound $w(J')$ only in terms of $H^+$ and $H^-$. This is proved in the next lemma.

**Lemma 4.6 restated.** *If $|S| > 2$, we have*

$$w(J') \le 3w(H^+) + 5w(H^-).$$

Note that, when $|S| > 2$, we have $T \setminus X^* \ne \emptyset$ and at least one of $H^+$ and $H^-$ is non-empty. We use the following notation. Fix $e^* \in X^*$ such that $w(e^*)$ is minimal over the edges in $X^*$ and denote $Y^* = X^* \setminus \{e^*\}$. We will bound the weight of the $S$-join $J' = X^* \Delta T$ piece by piece, as visualized in Figure 8. The weight of $e^*$ will be bounded by case analysis.

*Observation* A.4. If $|S| > 2$ and $e^* \in T$, we have $|Y^* \setminus T| \ge 2$.

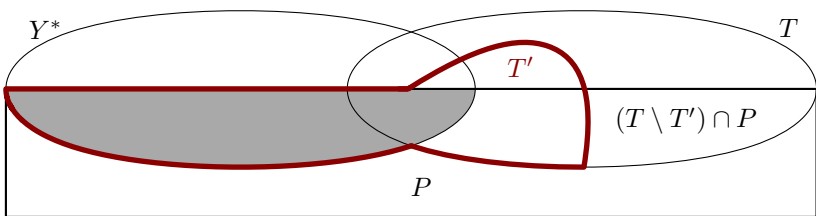

*Figure 9.* $T$ and the new spanning tree $T'$ containing $Y^* \cap P$.

*Proof.* The only spanning trees contained in the tour $X^*$ are paths, i.e., they have only two odd vertices. Therefore, if $|S| > 2$, there is $e \in T \setminus X^*$. Since $e^* \in T$, we have $\{e, e^*\} \subseteq T \setminus Y^*$. Finally, observe that $|Y^*| = |T|$, hence $|Y^* \setminus T| = |T \setminus Y^*| \geq 2$. □

*Observation* A.5. We have

1. $w(T \setminus X^*) \leq w(H^+) + w(H^-)$

2. $w((X^* \setminus T) \setminus P) \leq w(H^-)$.

*Proof.* The first statement follows from $T \setminus X^* = ((T \setminus X^*) \cap P) \cup ((T \setminus X^*) \setminus P) \subseteq (P \setminus X^*) \cup (T \setminus P)$, the definition of $H^+$ and Lemma A.2.

The second statement follows from $(X^* \setminus T) \setminus P \subseteq X^* \setminus P = H^-$. □

*Observation* A.6. We have
$$w((Y^* \setminus T) \cap P) \leq w(T \setminus Y^*).$$

*Proof.* Consider the spanning tree $T'$ created by adding some edges of $T$ to $Y^* \cap P$ (which is a forest). This way, $T' \setminus T = (Y^* \cap P) \setminus T = (((Y^* \cap T) \cap P) \cup ((Y^* \setminus T) \cap P)) \setminus T = (Y^* \setminus T) \cap P$, see also Figure 9.

Therefore, it is enough to bound the weight of $T' \setminus T$. By the minimality of $T$, we have $w'(T') \geq w'(T)$ and therefore $w'(T' \setminus T) \geq w'(T \setminus T')$ by subtracting $w'(T' \cap T)$ on both sides. Since $T' \setminus T \subseteq P$,

$$w(T' \setminus T) = -w'(T' \setminus T) \leq -w'(T \setminus T') = w((T \setminus T') \cap P) - w((T \setminus T') \setminus P) \leq w(T \setminus Y^*).$$

The last inequality follows from $w$ being non-negative and $(T \setminus T') \cap P \subseteq T \setminus Y^*$, see Figure 9. This concludes the proof, since $T' \setminus T = (Y^* \setminus T) \cap P$. □

*Proof of Lemma 4.6.* We consider the following cases:

**Case $e^* \notin P$.** Here, $e^* \in H^-$ and therefore $w(e^*) \leq w(H^-)$. Therefore, we have

$$w((X^* \setminus T) \cap P) = w((Y^* \setminus T) \cap P) \overset{Obs.\ A.6}{\leq} w(T \setminus Y^*) \leq w(T \setminus X^*) + w(e^*) \leq w(T \setminus X^*) + w(H^-). \quad (4)$$

We get

$$\begin{aligned}
w(J') &= w((X^* \setminus T) \setminus P) + w((X^* \setminus T) \cap P) + w(T \setminus X^*) \\
&\overset{Obs.\ A.5.(2)}{\leq} w(H^-) + w((X^* \setminus T) \cap P) + w(T \setminus X^*) \\
&\overset{(4)}{\leq} w(H^-) + (w(T \setminus X^*) + w(H^-)) + w(T \setminus X^*) \\
&\overset{Obs.\ A.5.(1)}{\leq} 2w(H^+) + 4w(H^-).
\end{aligned}$$

which implies the statement of the lemma.

**Case** $e^* \in P \cap T$. We have $X^* \setminus T = Y^* \setminus T$ and $T \setminus Y^* = (T \setminus X^*) \cup \{e^*\}$. We have

$$w(X^* \setminus T) = w((X^* \setminus T) \setminus P) + w((Y^* \setminus T) \cap P) \overset{Obs.\ A.6}{\leq} w(H^-) + w(T \setminus Y^*) = w(H^-) + w(T \setminus X^*) + w(e^*).$$

Since $e^*$ is the lightest edge in $X^*$ and $|X^* \setminus T| = |Y^* \setminus T| \geq 2$ (Observation A.4), we have $w(e^*) \leq \frac{1}{2}w(X^* \setminus T)$. Therefore the previous equation implies

$$w(X^* \setminus T) \leq 2(w(X^* \setminus T) - w(e^*)) \leq 2(w(H^-) + w(T \setminus X^*)).$$

We get

$$w(J') = w(X^* \setminus T) + w(T \setminus X^*) \leq 2w(H^-) + 3w(T \setminus X^*) \overset{Obs.\ A.5.(1)}{\leq} 3w(H^+) + 5w(H^-)$$

which implies the statement of the lemma.

**Case** $e^* \in P \setminus T$. First, we show that $w(e^*) \leq w(T \setminus X^*)$. Since $e^* \notin T$, there is a unique cycle $C \subseteq T \cup \{e^*\}$ and, since $T$ is an MST w.r.t. weights $w'$, all the edges $e \in C$ have weight $w'(e) \leq w'(e^*)$. Note that $C \setminus X^* \neq \emptyset$. Indeed, both $C$ and $X^*$ are cycles and $|X^*| = n \geq |C|$, hence the only possibility for $C \setminus X^* = \emptyset$ is that $C = X^*$. However in the latter case $|S| = |\text{odd}(T)| = 2$, a contradiction. Thus, we can choose $f \in C \setminus X^*$ and we have $w'(f) \leq w'(e^*) < 0$ (because $e^* \in P$), hence $w(T \setminus X^*) \geq w(f) \geq w(e^*)$. Therefore, we have

$$w((X^* \setminus T) \cap P) = w((Y^* \setminus T) \cap P) + w(e^*) \overset{Obs.\ A.6}{\leq} w(T \setminus Y^*) + w(e^*)$$

$$\leq 2w(T \setminus X^*) \overset{Obs.\ A.5.(1)}{\leq} 2w(H^+) + 2w(H^-). \tag{5}$$

The second last inequality follows from $T \setminus Y^* = T \setminus X^*$ (since $e^* \notin T$) and $w(e^*) \leq w(T \setminus X^*)$. Thus

$$\begin{aligned}
w(J') &= w((X^* \setminus T) \setminus P) + w((X^* \setminus T) \cap P) + w(T \setminus X^*) \\
&\overset{Obs.\ A.5.(2)}{\leq} w(H^-) + w((X^* \setminus T) \cap P) + w(T \setminus X^*) \\
&\overset{(5)}{\leq} w(H^-) + 2w(H^+) + 2w(H^-) + w(T \setminus X^*) \\
&\overset{Obs.\ A.5.(1)}{\leq} 3w(H^+) + 4w(H^-).
\end{aligned}$$

This concludes the proof of the lemma. $\qquad\square$

*Proof of Theorem 1.2.* Algorithm 2 returns a shortcutting of the multiset $T + J$ whose weight is, by triangle inequality, at most $w(T) + w(J)$.

In case $|S| = 2$, $J$ is the optimal T-join of the odd vertices in $T$, i.e, $w(J) \leq w(J')$. Combining Lemma A.2 and Lemma A.3 we get

$$\begin{aligned}
w(T) + w(J) &\leq w(T) + w(J') \\
&\leq w(T \cap X^*) + w(H^+) + w(H^-) + w(X^* \setminus T) + w(H^+) + w(H^-) \\
&\leq w(X^*) + 2w(H^+) + 2w(H^-).
\end{aligned}$$

If $|S| > 2$, $J$ is a 2-approximation of the optimal T-join, i.e., $w(J) \leq 2w(J')$. In this case, we use Lemma A.2 and Lemma 4.6:

$$\begin{aligned}
w(T) + w(J) &\leq w(T) + 2w(J') \\
&\leq w(T \cap X^*) + w(H^+) + w(H^-) + 2\big(3w(H^+) + 5w(H^-)\big) \\
&\leq w(X^*) + 7w(H^+) + 11w(H^-). \qquad\square
\end{aligned}$$

# B. Euclidean TSP

Here we prove Proposition 5.1. Delaunay triangulation is a classical object in Computational Geometry, see (Berg et al., 2008, Chapter 9). It can be defined as any triangulation of the Delaunay Graph which has the following characterization.

**Proposition B.1** (see Theorem 9.6 in (Berg et al., 2008)). *Let $V$ be a set of points in the plane. Two points $u, v \in V$ form an edge of the Delaunay graph of $V$ if and only if there is a closed disk containing $u$ and $v$ on its boundary which does not contain any other point of $V$.*

The proof of the following proposition is a variation of the classical proof that a Euclidean MST of a set of points is a subgraph of its Delaunay triangulation.

**Proposition 5.1 (restated).** *Let $V$ be a set of $n$ points in the Euclidean plane, $G = (V, \binom{V}{2})$ and $P \subseteq \binom{V}{2}$ with weights $w'$ such that $w'(uv) < 0$ for every $uv \in P$ and $w'(uv) = \|u - v\|_2$ for every $e \in E \setminus P$. Let $D \subseteq \binom{V}{2}$ be the Delaunay triangulation of $V$. Then $D \cup P$ contains an MST $T$ with respect to $w'$.*

*Proof.* Consider a fixed MST $T$ and any edge $uv = e \in T$. If $e \in P \cup D$, we are done. Otherwise, we show that $T$ is not an MST.

By Proposition B.1, every closed disc having $u, v$ on its boundary contains another point of $V$, since $uv \notin D$ and therefore it is not contained in the Delaunay Graph of $V$. Consider a closed disc $D$ whose diameter is the segment $uv$ and let $z \in V \setminus \{u, v\}$ be another point contained in $D$. We have $\|z - v\|_2 < \|u - v\|_2$ and $\|z - u\|_2 < \|u - v\|_2$. Therefore we have $w'(uz) < w'(uv)$ and $w'(vz) < w'(uv)$ regardless whether $uz$ and $vz$ belong to $P$, since the weights of edges in $P$ are negative. Without loss of generality, assume that $z$ and $u$ are in the same component of $T \setminus uv$. Since $w'(vz) < w'(uv)$, $(T \setminus \{uv\}) \cup \{vz\}$ is a spanning tree of a smaller weight than $T$. □

# C. Algorithms for Graphical TSP

In case of graphical TSP, the optimal tour may use some edges several times. Therefore, we will denote $\bar{X}^*$ the multiset of edges, where each edge is contained in $\bar{X}^*$ as many times as it is traversed by the optimal tour. On the other hand, we denote $X^*$ the ground set of $\bar{X}^*$ containing all edges which are traversed by the optimal tour at least once.

### C.1. Analysis of Algorithm 3: near-linear time algorithm for graphical TSP

We provide full analysis of Algorithm 3 proposed in Section 6. Similarly to Algorithm 2, it computes a 2-approximate $\text{odd}(T)$-join using the algorithm by Goemans & Williamson (1995) which runs in time $O(m \log n)$, where $m = |E|$ is the number of edges in the input graph. However, to achieve $O(\eta)$ additive error, it needs to receive a more verbose prediction in input. In particular, we assume that the prediction $P$ is a multiset and, if correct, contains each edge with the same multiplicity as the optimal TSP tour $\bar{X}^*$.

---

**Algorithm 3:** Almost-linear time algorithm for arbitrary graphs (restated for convenience)

**Input:** graph $G = (V, E)$, weights $w \colon E \to \mathbb{R}_+$, predicted multiset $P$ of edges in $E$
**Output:** TSP tour over $G$

**for** $e \in E$ **do** $w'(e) := 0$ if $e \in P$ and $w'(e) := w(e)$ otherwise.
find $T :=$ minimum spanning tree on $G$ w.r.t. $w'$
$S :=$ set of odd vertices in $T$
**for** $e \in E$ **do** $w''(e) := 0$ if $e \in (P - T)$ and $w'(e) := w(e)$ otherwise.
$J :=$ 2-approximate $\text{odd}(T)$-join w.r.t. to $w''$
output a short-cutting of the Eulerian multiset $T + J$

---

**Lemma C.1.**
$$w(T) \leq w(T \cap X^*) + \eta.$$

*Proof.* Note that $P \cup H^- \supseteq X^*$ is a connected graph and its modified weight is equal to $w'(P) + w'(H^-) = w'(H^-) = w(H^-)$ because $w'(e) = 0$ for any $e \in P$. Since $T$ is an MST w.r.t. $w'$, we have $w'(T) \leq w'(P \cup H^-) \leq w(H^-)$. We

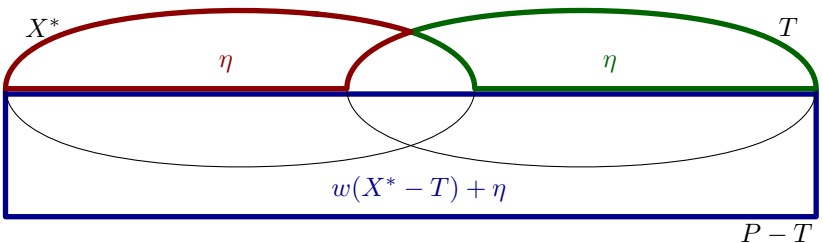

*Figure 10.* Parts of $J'$ and the bounds on their weight.

can write

$$w(T) = w(T \cap P) + w(T \setminus P) \le w(X^* \cap T) + w(H^+) + w(H^-), \tag{6}$$

where the inequality follows from $T \cap P \subseteq (T \cap X^*) \cup (T \cap H^+) \subseteq (T \cap X^*) \cup H^+$ and $w(T \setminus P) = w'(T) \le w(H^-)$. $\square$

We will also consider an auxiliary $\mathrm{odd}(T)$-join $J' = (T \setminus \bar{X}^*) + (\bar{X}^* - T)$. In the following observations, we bound the weight of its parts as in Figure 10.

*Observation* C.2. We have

$$w''(\bar{X}^* - T) = w((\bar{X}^* - T) - (P - T)) \le w(H^-).$$

*Proof.* We claim that

$$X^* - T \subseteq (P - T) + H^-. \tag{7}$$

To see this inclusion, we write the multiplicity of every edge $e \in X^* - T$ in the left-hand side of (7) as

$$\min\{m_{X^*}(e) - m_T(e), m_P(e)\} \le m_P(e) + m_{H^-} - m_T(e) \le m_{P-T}(e) + m_{H^-}.$$

Now, having proved (7), we can write

$$w((X^* - T) - (P - T)) = w''(X^* - T) \le w''(P - T) + w''(H^-) \le w(H^-),$$

because $w''(P - T) = 0$. $\square$

*Observation* C.3. We have

$$w''(T - X^*) = w((T - X^*) - (P - T)) \le \eta.$$

*Proof.* We can write $T = (T \cap X^*) + (T - X^*)$. Lemma C.1 then implies $w(T - X^*) \le \eta$. $\square$

*Observation* C.4. We have

$$w(P - T) \le w(X^* - T) + w(H^+).$$

*Proof.* We can write

$$P - T = ((P \cap X^*) + (P - X^*)) - T \subseteq ((P \cap X^*) - T) + (P - X^*) \subseteq (X^* - T) + H^+.$$

Therefore, we have $w(P - T) \le w(X^* - T) + w(H^+)$. $\square$

**Theorem 1.3 restated.** *Let $G = (V, E)$ be a graph on $n$ vertices and $m$ edges with weights $w \colon E \to \mathbb{R}_+$. Let $X^*$ be a TSP tour on $G$ and $P$ a prediction with error $\eta$ with respect to $X^*$. Then the cost of the tour $X$ produced by Algorithm 3 on $G$ with prediction $P$ is at most $w(X^*) + 4\eta$.*

*Proof.* We know that $J$ is a $S$-join, $S = \mathrm{odd}(T)$, such that $w''(J) \le 2w''(J')$ for any $S$-join $J'$. We bound the weight of $J$ by exposing a $S$-join $J'$ with a small weight. First, we show that $J' = (T - \bar{X}^*) + (\bar{X}^* - T)$ is a $S$-join. We can write

$$\begin{aligned}
T + J' &= ((T - \bar{X}^*) + (T \cap \bar{X}^*)) + ((T - \bar{X}^*) + (\bar{X}^* - T)) \\
&= ((T - \bar{X}^*) + (T - \bar{X}^*)) + ((T \cap \bar{X}^*) + (\bar{X}^* - T)) \\
&= ((T - \bar{X}^*) + (T - \bar{X}^*)) + \bar{X}^*.
\end{aligned}$$

Since $\bar{X}^*$ has all degrees even and each edge in $T - \bar{X}^*$ is doubled, we have that all the degrees in $T + J'$ are even, hence $J'$ is an $S$-join.

The weight of the resulting TSP tour is at most

$$w(T) + w(J) \le w(T) + w''(J) + w(P - T) \le (w(X^* \cap T) + \eta) + 2w''(J') + w(P - T), \tag{8}$$

where the first inequality holds because $J$ is a set and $w(J) = w''(J) + w(J \cap (P - T))$, and the last inequality follows from Lemma C.1 and the approximation guarantee on $J$. By Observations C.2 and C.3, we have

$$w''(J') = w''(T - \bar{X}^*) + w''(\bar{X}^* - T) \le \eta + w(H^-).$$

By Observation C.4, we have $w(P - T) \le w(X^* - T) + w(H^+)$. Summarizing, we get

$$w(T) + w(J) \le \big(w(X^* \cap T) + \eta\big) + 2\big(\eta + w(H^-)\big) + w(X^* - T) + w(H^+) \le w(X^*) + 4\eta. \qquad \square$$

## C.2. Analysis of Algorithm 1 in the setting of graphical TSP

We analyze Algorithm 1 in the graphical TSP setting and show that it satisfies the same guarantees as in metric TSP setting, i.e., Theorem 1. Compared to Algorithm 3, it does not require predicting multiplicities of edges in the optimal tour $\bar{X}^*$. Algorithm 1 receives only a prediction $P$ of the support set $X^*$ of edges used by $\bar{X}^*$ at least once. On the other hand, the running time of Algorithm 1 is $O(n^3)$ compared to the near-linear running time of Algorithm 3.

---

**Algorithm 1:** Restated for the graphical TSP setting

---

**Input:** graph $G = (V, E)$, weights $w \colon E \to \mathbb{R}_+$, prediction $P \subseteq E$
**Output:** TSP tour over $G$

**for** $e \in E$ **do** $w'(e) := 0$ if $e \in P$ and $w'(e) := w(e)$ otherwise.
find $T :=$ minimum spanning tree on $G$ w.r.t. modified weights $w'$
$S :=$ set of odd vertices in $T$
$J :=$ $S$-join of $S$ with minimum weight w.r.t. to the original weights $w$
output a shortcutting of the Eulerian multiset $T + J$

---

**Notation:** For an arbitrary optimal tour $\bar{X}^*$, we denote by $X^* \subseteq E$ the support of $\bar{X}^*$, i.e., the set of edges appearing in $\bar{X}^*$ at least once. Notice that $X^*$ is connected and $\bar{X}^*$ is Eulerian. Given a prediction $P \subseteq E$, we write $P = (X^* \setminus H^-) \cup H^+$, where $H^-$ and $H^+$ are the sets of false negatives and false positives. Notice that $\eta = w(H^-) + w(H^+)$.

**Theorem C.5.** *Let $G = (V, E)$ be a graph on $n$ vertices with edge weights $w \colon E \to \mathbb{R}_+$. There is an algorithm running in time $O(n^3)$ which, receiving a prediction $P \subseteq E$ with error $\eta$ with respect to some TSP tour $X^*$, finds a TSP tour $X$ of weight*

$$w(X) \le w(X^*) + 2\eta.$$

*Proof.* We will show that $w(T) + w(J) \le w(\bar{X}^*) + 2w(H^-) + 2w(H^+)$. By triangle inequality, the weight of the final shortcutting can be only smaller.

We start by bounding $w(T)$. First, note that $P \cup H^- \supseteq X^*$ is a connected graph and its modified weight is equal to $w'(P) + w'(H^-) = w'(H^-) = w(H^-)$ because $w'(e) = 0$ for any $e \in P$. Since $T$ is an MST w.r.t. $w'$, we have $w'(T) \le w'(P \cup H^-) \le w(H^-)$. We can write

$$w(T) = w(T \cap P) + w(T \setminus P) \le w(X^* \cap T) + w(H^+) + w(H^-), \tag{9}$$

where the inequality follows from $T \cap P \subseteq (T \cap X^*) \cup (T \cap H^+) \subseteq (T \cap X^*) \cup H^+$ and

$$w(T \setminus P) = w'(T) \le w(H^-). \tag{10}$$

Now, we bound $w(J)$. Instead of $J$, we consider the weight of $J' := (\bar{X}^* - T) + (T \setminus X^*)$ which is also a $S$-join, $S = \text{odd}(T)$, and therefore $w(J) \le w(J')$ by the optimality of $J$. Notice that $\bar{X}^* - T$ contains all the edges in $\bar{X}^*$ not

contained in $T$ with their multiplicity, and all the edges in $\hat{X}^*$ with multiplicity higher than 1 which are also contained in $T$ with their multiplicity decreased by 1. In particular, we have $\bar{X}^* = (\bar{X}^* - T) + (X^* \cap T)$. To show that $J'$ is an $S$-join, we equivalently prove that that $T + J'$ has all degrees even. We can write

$$T + J' = \big((T \cap X^*) + (T \setminus X^*)\big) + \big((\bar{X}^* - T) + (T \setminus X^*)\big) = \bar{X}^* + \big((T \setminus X^*) + (T \setminus X^*)\big),$$

because $(T \cap X^*) + (\bar{X}^* - T) = \bar{X}^*$. In other words, $T + J'$ is a sum of the tour $\bar{X}^*$, that has all even degrees, and of $(T \setminus X^*) + (T \setminus X^*)$, where each edge is taken twice.

We have $w(J') = w(\bar{X}^* - T) + w(T \setminus X^*)$, where

$$w(T \setminus X^*) = w((T \setminus X^*) \cap P) + w((T \setminus X^*) \setminus P) \leq w(H^+) + w(T \setminus P).$$

The last inequality follows from $(T \setminus X^*) \cap P \subseteq P \setminus X^* = H^+$ and from $(T \setminus X^*) \setminus P \subseteq T \setminus P$. We have

$$w(J) \leq w(J') = w(\bar{X}^* - T) + w(T \setminus X^*) \leq w(\bar{X}^* - T) + w(H^+) + w(T \setminus P) \overset{(10)}{\leq} w(\bar{X}^* - T) + w(H^+) + w(H^-). \tag{11}$$

Since $\bar{X}^* = (X^* \cap T) + (\bar{X}^* - T)$, equations (9) and (11) imply

$$w(T) + w(J) \leq w(\bar{X}^*) + 2w(H^+) + 2w(H^-). \qquad \square$$

## D. Lower Bounds and Counterexamples

Here we describe construction showing the tightness of our results and our analysis.

### D.1. Linear dependence on $\eta$

Algorithm 1 has approximation ratio $1 + 2\frac{\eta}{\text{OPT}}$ where OPT denotes the cost of an optimal solution. We next show that a linear dependence on $\eta/\text{OPT}$ is needed (though possibly with a factor smaller than the 2 that we get). The proof idea is that otherwise one could use such a learning-augmented algorithm for TSP to develop a too good approximation algorithm for TSP (without predictions).

**Theorem D.1.** *Assuming P$\neq$NP, there exists no polynomial-time learning-augmented algorithm for TSP with approximation factor smaller than $1 + \frac{1}{122}\frac{\eta}{\text{OPT}}$.*

*Proof.* Let us assume by contradiction that we are given a learning-augmented algorithm for TSP with approximation factor $1 + \alpha\frac{\eta}{\text{OPT}}$, for some constant $\alpha \in (0, \frac{1}{122})$. We show how to use it to construct a polynomial-time algorithm for TSP (without predictions) with approximation factor strictly smaller than $\frac{123}{122}$. Assuming P$\neq$NP, the latter algorithm is not possible (Karpinski et al., 2015).

Let $G$ be an input instance of TSP (with weights $w$), with optimal value $\text{OPT}_G$. The TSP algorithm (without predictions) works as follows. It guesses a value $X$ so that $\text{OPT}_G \leq X \leq (1 + \varepsilon)\text{OPT}_G$, where $\varepsilon > 0$ is a sufficiently small constant depending on $\frac{1}{122} - \alpha$. This can be done by, e.g., computing a 2-approximation $X'$ of $\text{OPT}_G$, and then considering as candidate values of $X$ all the powers of $(1 + \varepsilon)$ between $X'/2$ and $(1 + \varepsilon)X'$. At least one such value of $X$ satisfies the claim, and we run the rest of the algorithm for all the candidate $X$ (returning at the end the best solution obtained). Let $Y := \frac{1 - \alpha}{\alpha}X$. Then the algorithm picks an arbitrary node $v_0 \in G$, and creates a new graph $G'$ by adding to $G$ a new node $s$ and edges between $s$ and all other nodes with weights $w(s, u) = \frac{Y}{2} + w(v_0, u)$. It is easy to check that the optimal value $\text{OPT}_{G'}$ of the TSP instance on $G'$ (with weights $w$) satisfies $X + Y \geq \text{OPT}_{G'} \geq \frac{1}{1+\varepsilon}(X + Y)$. Maybe more details? Then the algorithm guesses what is the next node after $v_0$ in an optimal TSP tour in $G$ (there are $n - 1$ choices to check), let's call that guess $v_1$. The algorithm creates a prediction consisting of only two edges $(v_0, s)$ and $(s, v_1)$, and calls the hypothesized learning-augmented algorithm on $G'$ and that prediction. We claim that, in the iteration in which the guess of $v_1$ was correct, the prediction error is $\eta \leq X - w(v_0, v_1) \leqslant X$. Therefore the learning-augmented algorithm returns a solution of cost $APX_{G'} \leq \text{OPT}_{G'} + \alpha X$. By short-cutting node $s$ from that solution we obtain a solution for $G$ of cost at most

$$APX_{G'} - Y \leq \text{OPT}_{G'} + \alpha X - Y \leq (1 + \alpha)X \leq (1 + \varepsilon)(1 + \alpha)\text{OPT}_G.$$

Altogether we obtain a polynomial-time algorithm for TSP (without predictions) with approximation factor $(1 + \varepsilon)(1 + \alpha) < \frac{123}{122}$ for a sufficiently small constant $\varepsilon > 0$, a contradiction. $\qquad \square$

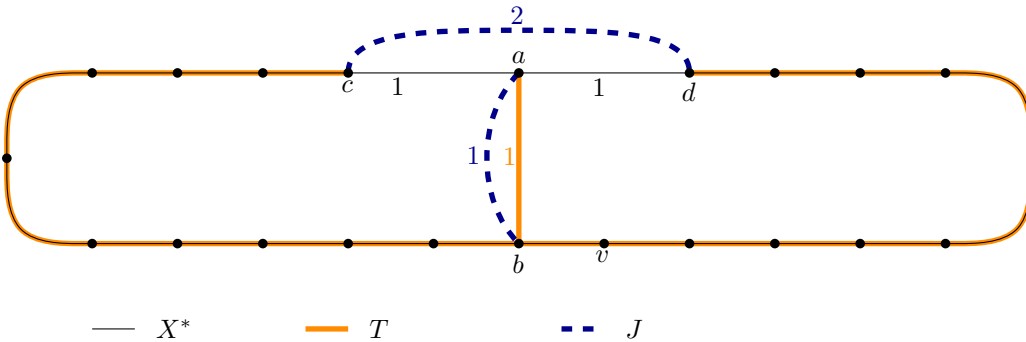

*Figure 11.* Tightness of our analysis with respect to the dependence on false positives. Path from $b$ to $c$ as well as the path from $b$ to $d$ has weight $M \gg 1$. The prediction $P = X^* \cup \{ab\}$, i.e., there is only one false positive $H^+ = \{ab\}$ and we have $\eta = w(ab) = 1$.

We remark that by a similar construction as in the proof of Theorem D.1, one can argue that a substantial improvement of the dependence on $\eta/\operatorname{OPT}$ in the approximation factor is very difficult. For example, a learning-augmented algorithm for TSP with approximation factor $1 + 0.49\frac{\eta}{\operatorname{OPT}}$ would imply a $1.49 + O(\varepsilon)$ approximation for TSP. The latter result would be a breakthrough in approximation algorithms given that only recently the 1.5 barrier was overcome and the best known algorithm has approximation ratio $1.5 - 10^{-36}$ (Karlin et al., 2021).

### D.2. Tightness of our analysis

Here we show that our analysis of Algorithm 1 is tight, in the sense that the coefficient 2 in front of the prediction error $\eta$ cannot be improved for the given algorithm. We provide two examples: one with false positives and one with false negatives. These two examples can be combined in a single input instance with the desired weight of false positives and false negatives. Note that Algorithm 1 satisfies the same approximation guarantees in both metric TSP and graphical TSP setting. In what follows, we first describe an input instance of graphical TSP which, by taking the metric closure of the proposed graph, gives an input instance of metric TSP. We describe the final outputs of Algorithm 1 in both setting, as they will differ slightly in the final short-cutting step.

**Prediction with false positives.** Figure 11 describes a bad example for the dependence on false positives. Given prediction $P = X^* \cup \{ab\}$, the set of thick orange edges is an MST with respect to the weights $w'$ in Algorithm 1. In the following steps, Algorithm 1 identifies $S = \{a, b, c, d\}$ as the set of odd vertices and the edges $J$ drawn with dark-blue dashed lines denote a min-weight $S$-join.

After short-cutting, the resulting tour in the graphical TSP setting will take the edge $ab$, then the path from $b$ to $c$ of length $M$, then taking edge $cd$, then path from $d$ to $b$ of length $M$, and then again edge $ab$.

In the metric TSP setting, we take the edge $ab$, then the path from $b$ to $c$ of length $M$, then taking edge $cd$, then path from $d$ to $v$ and finally the edge $va$. This forms a Hamiltonian cycle of the same weight as the graphical tour, since the weight of the edge $va$ in the metric closure is equal to the weight of the path $v, b, a$.

Therefore, in both cases the weight of the tour produced by Algorithm 1 will be

$$w(X) = 1 + M + 2 + M + 1 = 2M + 2 + 2 = w(X^*) + 2w(H^+) = w(X^*) + 2\eta,$$

showing that the approximation guarantee for Algorithm 1 as stated in theorems 1.1 and C.5 is tight.

**Prediction with false negatives.** Figure 12 describes a tight example for the dependence on false negatives. It shows an input where Algorithm 1 finds a tour with weight $w(X) \geq w(X^*) + 2\frac{k}{k+1}\eta$, where $2\frac{k}{k+1}$ can be arbitrarily close to 2 as $k$ can be chosen as large as roughly $n/10$. The graph $G$ in Figure 12 contains $k$ blocks, each containing 5 edges of length 1 and a single edge of length 2. The optimal tour $X^*$ uses 4 edges of unit length in each block and its total length is $M + 4k + 3$. The prediction error is $\eta = k + 1$: one edge of unit weight is missing in each block and another such edge on the right-hand side. On the other hand, Algorithm 1 finds an MST $T$ (fat orange) and an $S$-join for $S = \operatorname{odd}(T)$ (fat dashed

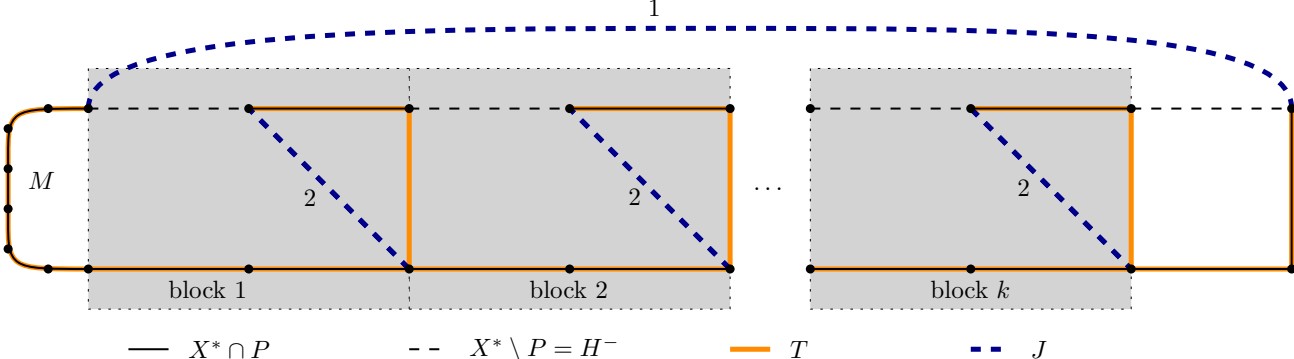

*Figure 12.* Tightness of our analysis with respect to the dependence on false negatives. The optimal tour $X^*$ contains solid and dashed black edges. The prediction $P$ contains only sold black edges, there are no false positives. Each of the $k$ blocks contains five edges of length 1 and one edge of length 2. The prediction error is $\eta = k + 1$.

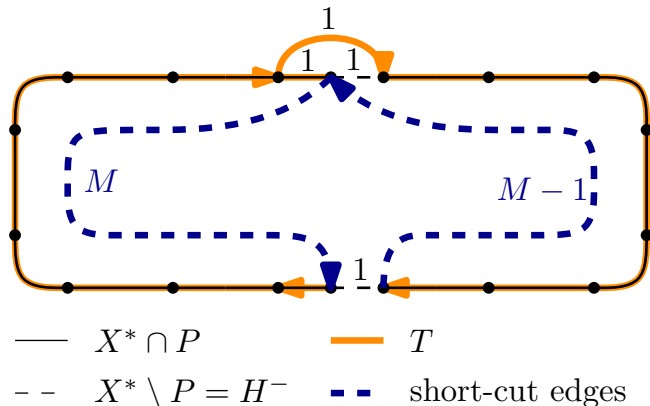

*Figure 13.* Bad example for the doubling-tree-inspired variant. Both the left-hand side and the right-hand side half of the optimal tour $X^*$ have weight $M$. The short-cutting edges have the same length as the shortest path between their endpoints.

blue), using altogether 4 edges of unit length and one edge of length 2 in each block. In total, it produces a tour of length

$$w(X) = M + k(4+2) + 2 + 1 = w(X^*) + 2k = w(X^*) + 2\frac{k}{k+1}\eta.$$

This shows that the approximation guarantee for Algorithm 1 as stated in theorems 1.1 and C.5 is tight up to the factor $\frac{k}{k+1}$ which can be arbitrarily close to 1 as $k$ increases.

## D.3. Justification of the steps performed by our algorithms

### D.3.1. DOUBLING $T$ INSTEAD OF $\mathrm{odd}(T)$-JOIN

The example in Figure 13 shows that if, in Algorithm 1, we take $J = T$ instead of $J = \mathrm{odd}(T)$-join which is easy to find and inspired by the classical minimum spanning tree heuristic for TSP, see (Traub & Vygen, 2024, Section 1.4), the resulting algorithm would not have smooth dependence on $\eta$. In particular, $\eta = 2$ (there are two false negative edges of weight 1) and the tour produced by the algorithm would have weight $w(X) = 2M + 2M - 3 = 2w(X^*) - 3$ can be chosen arbitrarily small.

### D.3.2. $J$ HAS TO BE COMPUTED WITH RESPECT TO THE ORIGINAL WEIGHTS

If we computed both $T$ and $J$ with respect to the weight $w'$, our algorithm would be equivalent to running Christofides algorithm with the weight of predicted edges set to zero, which is an approach proposed by Antoniadis et al. (2025) in their meta-algorithm. This, however does not work. Consider the example in Figure 13. There, the cheapest $\mathrm{odd}(T)$-join with respect to the weights $w'$ would be $T$ itself with $w'(T) = 0$. At the end, we would receive a tour $X$ with weight

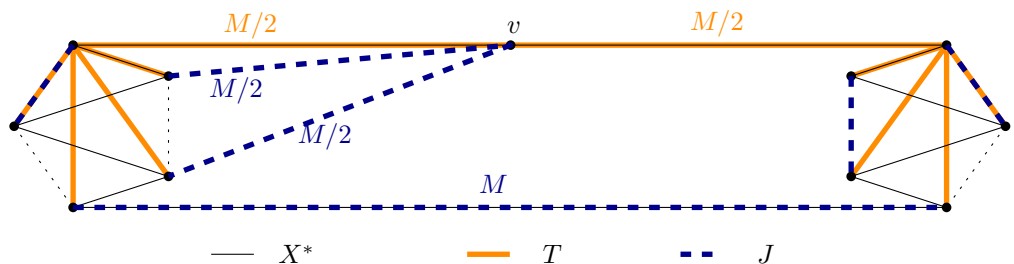

*Figure 14.* Example showing that Algorithm 1 replacing the optimal $\mathrm{odd}(T)$-join with a 2-approximation does not work. The graph $G$ consists of all edges drawn in the picture regardless of color and solid/dashed/dotted style. The edges of the two copies of $K_5$ on both sides have all weight 1. $P$ consists of all edges in the picture except for the two edges of length $M/2$ incident to $v$ marked with blue-dashed line, so $\eta = O(1)$.

$w(X) = 2w(X^*) - 3$ as argued in Section D.3.1.

### D.3.3. OPTIMAL $\mathrm{odd}(T)$-JOIN IN ALGORIHTM 1 CANNOT BE REPLACED BY A 2-APPROXIMATION

In Figure 14, we show an input graph which shows that Algorithm 1, if using a 2-approximate $\mathrm{odd}(T)$-join instead of the optimal one, would achieve approximation ratio close to 1.5 even with arbitrary small error. In the picture, $T$ is an MST and has five odd vertices in the left-hand side as well as in the right-hand side. Therefore, the optimal $\mathrm{odd}(T)$-join has to contain an edge (or a path) of length $M$ connecting one vertex in the left-hand side to another vertex in the right-hand side. The other odd vertices are then matched within their copy of $K_5$ in an arbitrary way. The total weight of the optimal $\mathrm{odd}(T)$-join is then $M + 4$. In Figure 14, there is an $\mathrm{odd}(T)$-join $J$ with weight $2M + 2$, i.e., it is a 2-approximate $\mathrm{odd}(T)$-join. A tour which starts at vertex $v$ and continues over one of the dashed-blue edges in $J$, even after short-cutting will have cost $3M + O(1)$, while the optimal tour has cost $2M + O(1)$.

### D.4. Non-metric TSP

It is well known that general TSP without the triangle inequality is inapproximable: unless P=NP, no polynomial-time algorithm can achieve any finite approximation ratio for non-metric TSP (Sahni & Gonzalez, 1976). Unfortunately, as we show below, this hardness persists even when predictions with arbitrarily small error are available.

Specifically, for every $r > 1$ and $\epsilon > 0$, we show that there is no polynomial-time $r$-approximation algorithm for non-metric TSP, even when given predictions of error at most $\epsilon$, unless P=NP. The proof is by reduction from Hamiltonian Cycle.

Without loss of generality, assume $\epsilon < 1/r$. Let $G = (V, E)$ be an instance of Hamiltonian Cycle on $n = |V|$ vertices. We construct an instance of non-metric TSP on the complete graph $K = \left(V, \binom{V}{2}\right)$ with edge weights defined by

$$w_e = \begin{cases} \epsilon/n & \text{if } e \in E, \\ 1 & \text{if } e \notin E. \end{cases}$$

We also provide the prediction $\hat{X} = \emptyset$.

If $G$ contains a Hamiltonian cycle, then the corresponding tour in $K$ uses only edges of weight $\epsilon/n$, and therefore has total length $\epsilon$. In this case, the prediction error is $\eta = \epsilon$.

On the other hand, any tour that uses at least one edge outside $E$ has cost strictly larger than 1. Since $\epsilon < 1/r$, we have $1 > r\epsilon$. Therefore, every non-optimal tour has length greater than $k$ times the optimum. Consequently, any $r$-approximation algorithm must return the optimal tour itself, which necessarily corresponds to a Hamiltonian cycle in $G$. Hence, such an algorithm would solve Hamiltonian Cycle in polynomial time, implying P=NP.

## E. More Details on the Experiments Provided in Section 7

In this section, we include additional explanations and comments on our experiments.

### E.1. Implementation details

The code was implemented in Python 3.13. For graph-related tasks, such as the computation of minimum spanning trees and minimum-weight $S$-joins, we used the `networkx` (Hagberg et al., 2008) package. We note that more efficient implementations exist for what is currently the main bottleneck of our procedure, namely, minimum-weight perfect matching, such as Blossom V (Kolmogorov, 2009). However, for this paper, we prioritized simplicity over efficiency. All experiments were performed on a cluster featuring dual Intel Xeon E5-2650 v3 CPUs with 128–512 GB of RAM. For neural network inference, we used an NVIDIA GeForce GTX 1080 GPU.

### E.2. On the score computation

In this section, we provide additional details on how the scores guiding the solution search methods are computed.

**DIFUSCO:** The paper (Sun & Yang, 2023) provides four checkpointed models, trained on instances with $n \in \{50, 100, 500, 1000\}$. For inference, we follow a policy of selecting the smallest checkpoint dimension that is greater than or equal to the target size for each problem size under consideration. When the instances are larger than 1000 nodes, we used the 1000 checkpoint.

**GNN-GLS:** The paper by (Hudson et al., 2022) proposed three pretrained models, on data of different sizes: $n_{\text{model}} \in \{20, 50, 100\}$. We adopt the following policy:

$$n_{\text{model}} = \begin{cases} 20 & n \leq 20 \\ 50 & 20 < n \leq 50 \\ 100 & \text{otherwise.} \end{cases}$$

The paper proposes to predict *regrets*, rather than probabilities, where a regret quantifies how undesirable it is to include a given edge in the candidate solution. We then convert regrets into probabilities through a nonincreasing transformation that incorporates a scaling component, rather than relying solely on normalization. The intuition is that regrets associated with "reasonable" edges (i.e., edges not trivially too long to be part of a good solution) should fall within the interval $\left[0, \frac{1}{n}\right]$. To enforce this behavior, each regret value $r_{ij}$ is mapped to a probability using

$$p_{ij} = \max(1 - n \cdot r_{ij}, 0), \tag{12}$$

which guarantees that larger regrets lead to smaller (and eventually zero) probabilities.

In cases where all predicted regrets map to zero, we default to a uniform distribution weighted by the inverse cost, namely,

$$p_{ij} = \frac{1}{w_{ij}}, \qquad \forall 1 \leq i < j \leq n.$$

**GNN4CO:** The paper by (Joshi et al., 2022) proposes two pre-trained models: one trained on instances of variable size and another trained on large cases with $n = 200$ nodes. They also introduce two decoding strategies: an autoregressive (AR) decoder and a non-autoregressive (NAR) decoder, where the edge probabilities are predicted simultaneously. In our experiments, we made the following choices. We use only the architecture trained on variable-size instances, as it exhibits better generalization properties (see Section 5.1 in (Joshi et al., 2022)). We adopt the AR decoding strategy, which yields better solution quality than the NAR decoder (see Figure 7 in (Joshi et al., 2022)). We use the same hyperparameter configuration as in the original paper and discard any predicted probabilities below 0.1, following their approach.

**SoftDist:** To obtain meaningful probability values, we first normalize the edge costs. For each graph, for each edge $e$, we define

$$\bar{w}(e) := \frac{w(e)}{\max_{e' \in E} w(w)},$$

so that all costs lie within the interval $[0, 1]$. To select the appropriate temperature, we choose the one corresponding to the closest instance size among those validated by Xia et al. (2024). For example, for an instance with 612 nodes, we use the temperature validated for $n = 500$. In the case of tiled size, we select the temperature associated with the largest number of nodes, as we observed a correlation between instance size and the lowest optimal temperature (e.g, for $n = 750$, we opt for the one associated to 1000).

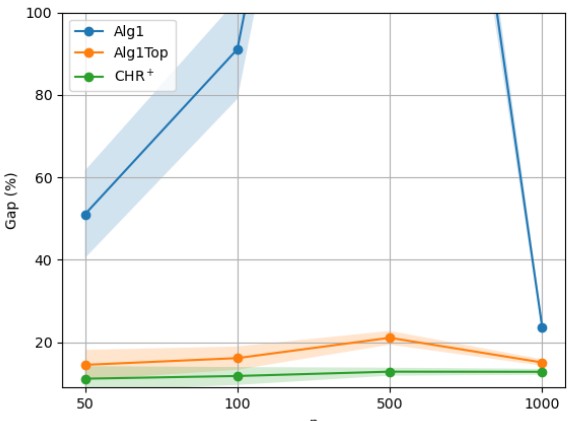

*Figure 15.* Comparative analysis of edge selection strategies using SoftDist predictions. We compare the optimality gaps of Alg1, Alg1Top, and CHR$^+$. We report the average optimality gap (percentage) on dataset $\mathcal{U}$.

**Synthetic predictors (SP)** Given $\varepsilon \in (0, 1)$ and letting $OPT$ denote the optimal tour value, let $X^*$ be the set of edges of an optimal tour. We sample a subset $H^- \subseteq X^*$ such that $\sum_{e \in H^-} w(e) \simeq \varepsilon \cdot OPT$ and define $H^+$ analogously. We then set $X' := (X^* \setminus H^-) \cup H^+$, and assign predictions equal to $p = 0.9$ to the edges in $X'$ and 0 otherwise.

For all predictors, we normalize the scores so that their maximum value is 1. An additional detail concerns normalization with respect to edge weights. For G1,G2, and BS we normalize the scores by the edge costs, as is standard in this line of work (see, e.g., (Sun & Yang, 2023)). For CHR$^+$, we do not apply this normalization. The reason is that our algorithm does not rely on heatmaps alone to construct a feasible solution, but explicitly incorporates the edge weights.

### E.3. Algorithm 1 vs CHR$^+$

A careful analysis of Algorithm 1 reveals that it requires a subset of edges $P$ (of cardinality $k$) whose associated weights are set to zero. In practice, given a heatmap of edge probabilities, several strategies can be employed to derive $P$. One approach is to sample $k$ edges according to the edge probability distribution (**Alg1**), while another is to select the $k$ edges with the highest probabilities (**Alg1Top**). Alternatively, rather than defining $w'(e)$ as in Algorithm 1, one can define the modified weight for every edge $e$ as $w'(e) = w(e)(1 - w(e))$ and execute Algorithm 1. Since this latter approach is essentially Christofides' algorithm applied to a modified graph, we denote it as **CHR$^+$**. While our theoretical analysis is developed for Alg1, we evaluate the three approaches. In all cases, we set $k = n$. Under the Alg1 approach, we generate 30 distinct edge sets $P$ for each instance to account for stochasticity; in contrast, the Alg1Top strategy defines $P$ deterministically based on the provided heatmap. Using SoftDist as a representative predictor, we compute the resulting optimality gaps for each strategy to assess their relative performance. Figure 15 demonstrates that CHR$^+$ consistently achieves superior performance. Consequently, we adopt CHR$^+$ as the primary decoding strategy for all subsequent experiments. Figure 15 demonstrates the empirical superiority of CHR$^+$ over Algorithm 1 in effectively leveraging neural predictions.

### E.4. Quality of the prediction error with respect to greedy

**G2 decoding has worse error dependency then CHR$^+$.** Figure 16 reports the average performance of Christofides, G2, and CHR$^+$ on the first five instances of our dataset with $n = 50$. The probabilities are generated using SP, and the sets $H^+$ and $H^-$ are sampled at random using 30 different seeds. Since the greedy approach performs significantly worse than the other methods, we exclude it from subsequent analyses. We report only G2 because our preliminary analysis indicates that G1 performs worse than G2.

**CHR$^+$ exhibits graceful degradation with respect to the prediction error.** Figure 5 shows how CHR$^+$ performances are with respect to the prediction error $\varepsilon$. To obtain this plot, we fix $n = 500$ and sample $H^+$ and $H^-$ at random using 5 different seeds.

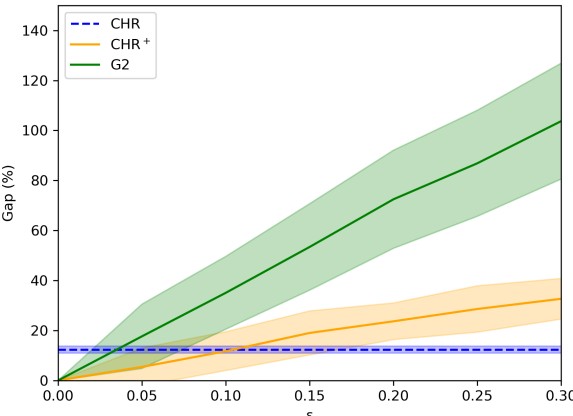

*Figure 16.* Average performance (Gap percentage) of Christofides, G2, and CHR$^+$ on the first five instances of our dataset with $n = 50$. The probabilities are generated using SP, and the sets $H^+$ and $H^-$ are sampled at random using 30 different seeds.

### E.5. Comparison of different solution search methods on dataset $\mathcal{U}$, $\mathcal{T}_E$, and $\mathcal{T}_M$

In this section, we report an extensive analysis of all the solution search methods on the dataset generated uniformly at random ($\mathcal{U}$) and the instances from TSPLIB (Euclidean, $\mathcal{T}_E$, metric non-Euclidean, $\mathcal{T}_M$). See Table 1, Table 2, and Table 3. For our experiments, the beam search width is set to 50, providing an optimal trade-off between solution quality and computational efficiency. The results on dataset $\mathcal{U}$ demonstrate that CHR$^+$ is the most robust decoding strategy as the instance size $n$ scales. While traditional solution search (G1, G2, BS) often sees its performance degrade significantly compared to the Christofides baseline as $n$ increases, CHR$^+$ consistently maintains or improves upon the baseline. Notably, for $n = 1000$, CHR$^+$ paired with SoftDist achieves a gap of 12.80%, outperforming plain Christofides (12.90%), whereas G1 and G2 degrade to 29.00% and 25.43% respectively. When paired with the DIFUSCO predictor, CHR$^+$ achieves state-of-the-art results on smaller instances, such as a 0.10% gap for $n = 50$. The empirical superiority of CHR$^+$ generalizes to the Euclidean TSPLIB benchmark (dataset $\mathcal{T}$). Across nearly all instances, CHR$^+$ produces the lowest integrality gaps. For example, on the pr76 instance, CHR$^+$ with DIFUSCO yields a gap of 1.91%, a significant improvement over the 7.88% gap provided by the standard Christofides algorithm. In the metric non-Euclidean setting, we can only evaluate SoftDist and GNNGLS, since the other two predictors are restricted to the Euclidean framework. In this regime, CHR$^+$ is overall less effective. Although the performance gains are less pronounced than in the Euclidean case – primarily because the problem is more challenging – and in some instances closer to those of the plain Christofides algorithm, our CHR$^+$ remains the most robust decoder across all instances. On average, it performs best when combined with GNNGLS, achieving an average gap of 8.49% compared to 8.70% for Christofides. This difference should be interpreted with some caution, as GNNGLS cannot compute heatmaps on large and difficult instances, which would likely increase the average gap.

### E.6. 2-opt Improvement

We evaluate the impact of 2-opt in CHR$^+$, G2, and BS. G1 is omitted due to its documented sub-optimality relative to G2; similarly, we restrict our predictors to SoftDist and DIFUSCO. The results for dataset $\mathcal{U}$ are summarized in Table 4. CHR$^+$ demonstrates superior performance across all dimensions except $n = 50$, where BS achieves a lower gap—primarily due to a superior initialization tour. Crucially, the number of 2-opt swaps required to reach local optimality is comparable between CHR$^+$ and the plain Christofides baseline. When considering the runtimes reported in Table 1, it is evident that CHR$^+$ has, on average, the same running time as Christofides. Consequently, the CHR$^+$ + 2-opt pipeline yields significantly higher solution quality than Christofides at an equivalent computational cost.

| Method | Dataset → Decoder | TSP50 Tour length | Gap (%) | Runtime (s) | TSP100 Tour length | Gap (%) | Runtime (s) | TSP500 Tour length | Gap (%) | Runtime (s) | TSP1000 Tour length | Gap (%) | Runtime (s) |
|---|---|---|---|---|---|---|---|---|---|---|---|---|---|
| Concorde | Exact | 5.69 | 0.00 | 0.06* | 7.76 | 0.00 | 0.24* | 16.55 | 0.00 | 18.67* | 23.12 | 0.00 | 84.41* |
| Christofides | Heuristic | 6.33 | 11.30 | 0.01 | 8.68 | 11.95 | 0.03 | 18.67 | 12.86 | 3.28 | 26.10 | 12.9 | 23.28 |
| SoftDist | G1 | 6.89 | 21.07 | 0.00 | 9.54 | 23.00 | 0.00 | 20.55 | 24.18 | 0.18 | 29.00 | 25.43 | 1.35 |
|  | G2 | 6.57 | 15.56 | 0.00 | 9.06 | 16.78 | 0.14 | 19.38 | 17.12 | 0.03 | 27.25 | 17.18 | 0.18 |
|  | BS | 6.94 | 22.01 | 0.03 | 9.99 | 28.78 | 0.14 | 22.68 | 37.11 | 9.46 | 30.93 | 33.77 | 71.92 |
|  | CHR$^+$ | 6.32 | 11.04 | 0.01 | 8.67 | 11.75 | 0.03 | **18.67** | **12.83** | 2.77 | **26.08** | **12.80** | 22.15 |
| DIFUSCO | G1 | 5.73 | 0.72 | 0.00 | 7.86 | 1.4 | 0.00 | 24.15 | 45.96 | 0.20 | 46.39 | 100.69 | 1.28 |
|  | G2 | 5.72 | 0.54 | 0.00 | 7.87 | 1.51 | 0.0 | 24.15 | 45.96 | 0.2 | 43.36 | 87.57 | 0.18 |
|  | BS | **5.70** | **0.14** | 0.03 | 7.79 | 0.48 | 0.14 | 29.11 | 75.94 | 10.35 | 52.01 | 125.0 | 68.85 |
|  | CHR$^+$ | 5.70 | 0.18 | 0.00 | **7.78** | **0.33** | 0.01 | 19.61 | 18.49 | 3.78 | 29.86 | 29.18 | 34.9 |
| GNNGLS | G1 | 6.90 | 21.37 | 0.00 | 10.6 | 36.66 | 0.00 | 30.57 | 84.75 | 0.18 | - | - | - |
|  | G2 | 6.33 | 11.18 | 0.00 | 9.27 | 19.58 | 0.00 | 24.98 | 50.96 | 0.05 | - | - | - |
|  | BS | 7.37 | 29.53 | 0.02 | 18.32 | 136.25 | 0.12 | 147.71 | 793.23 | 7.66 | - | - | - |
|  | CHR$^+$ | 6.33 | 11.23 | 0.01 | 8.68 | 11.92 | 0.04 | 18.67 | 12.87 | 2.85 | - | - | - |
| GNN4CO | G1 | 25.03 | 340.36 | 0.00 | 49.1 | 533.50 | 0.00 | 231.40 | 1298.68 | 0.18 | 452.02 | 1855.45 | 1.25 |
|  | G2 | 25.04 | 341.21 | 0.00 | 49.23 | 535.20 | 0.00 | 228.39 | 1280.47 | 0.05 | 452.02 | 1855.45 | 1.25 |
|  | BS | 24.53 | 331.91 | 0.03 | 48.86 | 530.48 | 0.13 | 231.68 | 1300.40 | 8.88 | 454.08 | 1864.37 | 63.29 |
|  | CHR$^+$ | 14.65 | 158.04 | 0.01 | 20.98 | 170.39 | 0.04 | 30.21 | 82.53 | 3.44 | 33.87 | 46.48 | 23.23 |

Table 1: Comparison of the performances of the four decoding strategies with different predictors on dataset $\mathcal{U}$ as taken from the paper (Ma et al., 2025b): results reported are the x mean of average tour length, optimality gap (percentage), and running time (seconds). Concorde running times, marked with (*), are taken from ML4CO-Bench-101. As the instance size increased, GNN-GLS became computationally intractable and was unable to produce valid heatmaps.

Table 2. Comparison of the performances of the four decoding strategies with different predictors on dataset $\mathcal{T}$: results reported are optimality gap (Percentage). As the instance size increased, GNN-GLS became computationally intractable and was unable to produce valid heatmaps.

| Algorithm ↓ | Decoder ↓ | Instance → | burma14 | ulysses16 | ulysses22 | bayg29 | att48 | gr96 | gr137 | si175 | gr202 | gr229 | gr431 | att532 | si535 | ali535 | gr666 | Avg. Vals ↓ |
|---|---|---|---|---|---|---|---|---|---|---|---|---|---|---|---|---|---|---|
| | $n$ | | 14 | 16 | 22 | 29 | 48 | 96 | 137 | 175 | 202 | 229 | 431 | 532 | 535 | 535 | 666 | |
| Concorde | | Opt. | 3323 | 6859 | 7013 | 1610 | 10628 | 55209 | 69853 | 21407 | 40160 | 134602 | 171414 | 27686 | 48450 | 202339 | 294358 | - |
| | | Runtime (s) | 0 | 0.02 | 0.04 | 0.01 | 0.06 | 0.23 | 0.24 | 0.89 | 0.35 | 3.91 | 8.91 | 9.61 | 2.44 | 2.28 | 7.45 | 2.43 |
| Christofides | | Tour length | 3606 | 6983 | 7411 | 1716 | 12613 | 61951 | 76589 | 21954 | 43377 | 147311 | 186132 | 31219 | 50255 | 225134 | 326784 | - |
| | | Gap (%) | 8.52 | 1.81 | 5.68 | **6.58** | 18.68 | 12.21 | **9.64** | 2.56 | **8.01** | **9.44** | **8.59** | 12.76 | 3.73 | **11.27** | 11.02 | 8.70 |
| | | Runtime (s) | 0.00 | 0.00 | 0.00 | 0.00 | 0.01 | 0.02 | 0.04 | 0.08 | 0.25 | 0.34 | 2.01 | 2.29 | 2.56 | 3.00 | 5.64 | 1.08 |
| SoftDist | G1 | Tour length | 4122 | 7179 | 7654 | 1874 | 12140 | 67063 | 89865 | 22306 | 48259 | 167203 | 214454 | 34227 | 49959 | 249587 | 352839 | - |
| | | Gap (%) | 24.04 | 4.67 | 9.14 | 16.40 | **14.23** | 21.47 | 28.65 | 4.20 | 20.17 | 24.22 | 25.11 | 23.63 | 3.11 | 23.35 | 19.87 | 17.48 |
| | | Runtime (s) | 0.00 | 0.00 | 0.00 | 0.00 | 0.00 | 0.00 | 0.00 | 0.01 | 0.01 | 0.02 | 0.11 | 0.20 | 0.21 | 0.21 | 0.39 | 0.08 |
| | G2 | Tour length | 3876 | 7771 | 7965 | 1831 | 13093 | 63278 | 80791 | 21925 | 46018 | 152241 | 194880 | 32554 | 49159 | 232568 | 335261 | - |
| | | Gap (%) | 16.64 | 13.30 | 13.57 | 13.73 | 23.19 | 14.62 | 14.92 | **2.42** | 14.59 | 13.10 | 13.69 | 17.58 | **1.46** | 14.94 | 13.90 | 13.44 |
| | | Runtime (s) | 0.00 | 0.00 | 0.00 | 0.00 | 0.00 | 0.00 | 0.00 | 0.00 | 0.01 | 0.00 | 0.02 | 0.04 | 0.03 | 0.03 | 0.05 | 0.01 |
| | BS | Tour length | 3953 | 6971 | 7278 | 2179 | 12377 | 65501 | 81968 | 22258 | 50205 | 176573 | 237226 | 38930 | 50332 | 276774 | 395286 | - |
| | | Gap (%) | 18.96 | 1.63 | **3.78** | 35.34 | 16.46 | 18.64 | 17.34 | 3.98 | 25.01 | 31.18 | 38.39 | 40.61 | 3.88 | 36.79 | 34.29 | 21.75 |
| | | Runtime (s) | 0.00 | 0.00 | 0.00 | 0.00 | 0.00 | 0.02 | 0.04 | 0.07 | 0.10 | 0.14 | 0.77 | 1.44 | 1.42 | 1.45 | 2.68 | 1.04 |
| | CHR⁺ | Tour length | 3750 | 7098 | 7411 | 1716 | 12486 | 61951 | 76589 | 21972 | 43377 | 150731 | 190024 | 31084 | 50050 | 227139 | 325623 | - |
| | | Gap(%) | 12.85 | 3.48 | 5.68 | **6.58** | 17.48 | 12.21 | **9.64** | 2.64 | **8.01** | 11.98 | 10.86 | **12.27** | 3.30 | 12.26 | **10.62** | 9.32 |
| | | Runtime (s) | 0.00 | 0.00 | 0.00 | 0.00 | 0.01 | 0.02 | 0.05 | 0.09 | 0.27 | 0.26 | 2.08 | 2.45 | 2.13 | 2.62 | 5.66 | 1.04 |
| GNNGLS | G1 | Tour length | 3998 | 6943 | 7408 | 2005 | 13698 | 76385 | 100791 | 22263 | 63113 | 181847 | 252664 | 35516 | 50145 | 733650 | - | - |
| | | Gap(%) | 20.31 | 1.22 | 5.63 | 24.53 | 28.89 | 38.36 | 44.29 | 4.00 | 57.15 | 35.10 | 47.40 | 28.28 | 3.50 | 262.58 | - | 43.19 |
| | | Runtime(s) | 0.00 | 0.00 | 0.00 | 0.00 | 0.00 | 0.00 | 0.00 | 0.01 | 0.01 | 0.02 | 0.11 | 0.20 | 0.22 | 0.21 | - | 0.06 |
| | G2 | Tour length | 3381 | 7989 | 7958 | 1974 | 13884 | 61482 | 84868 | 21957 | 57008 | 164788 | 235945 | 34002 | 49413 | 792445 | - | - |
| | | Gap (%) | **1.75** | 16.47 | 13.47 | 22.61 | 30.64 | **11.36** | 21.50 | 2.57 | 41.95 | 22.43 | 37.65 | 22.81 | 1.99 | 291.64 | - | 38.84 |
| | | Runtime (s) | 0.00 | 0.00 | 0.00 | 0.00 | 0.00 | 0.00 | 0.00 | 0.00 | 0.00 | 0.01 | 0.01 | 0.03 | 0.07 | 0.07 | 0.05 | 0.02 |
| | BS | Tour length | 3506 | 6900 | 7728 | 2186 | 12223 | 85666 | 110524 | 22447 | 67095 | 192674 | 237335 | 41957 | 50266 | 2877561 | - | - |
| | | Gap (%) | 5.51 | **0.60** | 10.20 | 35.78 | 15.01 | 55.17 | 58.22 | 4.86 | 67.07 | 43.14 | 38.47 | 51.55 | 3.75 | 1322.15 | - | 122.07 |
| | | Runtime (s) | 0.00 | 0.00 | 0.00 | 0.00 | 0.00 | 0.02 | 0.04 | 0.07 | 0.10 | 0.14 | 0.75 | 1.37 | 1.43 | 1.31 | - | 0.38 |
| | CHR⁺ | Tour length | 3606 | 6983 | 7411 | 1716 | 12520 | 61951 | 76589 | 21954 | 43377 | 147311 | 186132 | 31219 | 50255 | 225636 | - | - |
| | | Gap (%) | 8.52 | 1.81 | 5.68 | **6.58** | 17.80 | 12.21 | **9.64** | 2.56 | **8.01** | **9.44** | **8.59** | 12.76 | 3.73 | 11.51 | - | **8.49** |
| | | Runtime (s) | 0.00 | 0.00 | 0.00 | 0.00 | 0.01 | 0.03 | 0.05 | 0.19 | 0.27 | 0.27 | 2.19 | 2.63 | 2.69 | 2.98 | - | 0.80 |

*Table 3.* Results on non-Euclidean (metric) TSP instances for two predictors: GNNGLS and SoftDist.

| | $n \rightarrow$ | 50 | | 100 | | 500 | | 1000 | |
|---|---|---|---|---|---|---|---|---|---|
| | | Gap (%) | Swaps | Gap (%) | Swaps | Gap (%) | Swaps | Gap (%) | Swaps |
| | Christofides | 3.93 ± 2.18 | 10 ± 4 | 4.52 ± 1.70 | 20 ± 5 | 5.01 ± 0.78 | 104 ± 11 | 5.00 ± 0.56 | 207 ± 17 |
| CHR⁺ | SoftDist | 3.80 ± 2.19 | 10 ± 4 | 4.44 ± 1.68 | 20 ± 5 | **4.99 ± 0.77** | 104 ± 11 | **4.98 ± 0.56** | 206 ± 17 |
| | DIFUSCO | 0.10 ± 0.27 | 0 ± 1 | **0.23 ± 0.33** | 1 ± 1 | 5.65 ± 0.88 | 149 ± 15 | 7.26 ± 0.82 | 434 ± 33 |
| G2 | SoftDist | 3.95 ± 2.60 | 13 ± 6 | 4.74 ± 2.13 | 25 ± 8 | 5.19 ± 0.98 | 117 ± 20 | 5.36 ± 0.68 | 244 ± 30 |
| | DIFUSCO | 0.15 ± 0.48 | 1 ± 2 | 0.43 ± 0.93 | 3 ± 5 | 6.92 ± 1.30 | 238 ± 28 | 9.20 ± 0.95 | 1115 ± 52 |
| BS | SoftDist | 5.73 ± 3.35 | 15 ± 7 | 7.25 ± 2.63 | 36 ± 11 | 8.55 ± 1.27 | 219 ± 25 | 7.99 ± 0.89 | 396 ± 36 |
| | DIFUSCO | **0.09 ± 0.25** | 0 ± 1 | 0.25 ± 0.45 | 1 ± 2 | 8.68 ± 1.40 | 461 ± 43 | 9.18 ± 0.89 | 1448 ± 72 |

*Table 4.* Comparison of the performances of G2, BS, and CHR⁺ enhanced with the 2-opt swaps, with different predictors on dataset $\mathcal{U}$: Integrality gap (%) and number of 2-opt swaps (± standard deviation).

