# OpenReview forum: "TSP with Predictions: Heatmap to Tour with Provable Guarantees"
_ICML.cc/2026/Conference — ICML 2026 regular_

### Official Review · Reviewer_PtpL · 2026-02-19

**Soundness:** 3
**Presentation:** 3
**Significance:** 3
**Originality:** 3
**Overall Recommendation:** 5
**Confidence:** 4

**Summary:**

The paper studies learning-augmented approximation algorithms for the fundamental
Traveling Salesperson Problem (TSP).
Learning-augmented algorithms are a by now an established subfield of beyond worst-case analysis,
which received a lot of attention in recent years, both in ML and TCS conferences.

In the (metric) TSP problem, one is given a metric space with $n$ points. The goal is to find a tour of minimum length that visits each point exactly once. In terms of polynomial-time approximation algorithms,
this problem is APX-hard and the best-known approximation ratio is slightly below $3/2$.
This is one of the most fundamental problems in combinatorial optimization and operations research.

The authors study this problem in the context of learning-augmented algorithms,
where for each pair of points, a value between $0$ and $1$ is given which represents the likelihood that this edge is used in an optimal tour. One can interpret this information as a heatmap.
They give an algorithm that is inspired by the fundamental TSP algorithm by Christofides to compute
a solution with an approximation ratio of at most $1 + 2\eta/OPT$, where $\eta$ denotes the prediction error.
This value describes how accurate the prediction actually was compared to the closest optimal solution.
Formally, it is defined as the L1 norm between an optimal tour and the predicted values over all pairs of nodes. The authors also show that a linear dependence on $\eta$ is unavoidable.

Besides this main result, the paper contains faster and more practical algorithms with a slightly worse dependence on $\eta$, and variants for Graphical and Euclidean TSP. Finally, the paper implements
these algorithms, uses known predictors from Graph Neural Networks to generate heatmaps, and show a strong
performance of their algorithm compared to many baselines.

**Compliance With Llm Reviewing Policy:**

Affirmed.

**Final Justification:**

I keep my initial assessment and support the acceptance of this paper.

**Key Questions For Authors:**

Can you say somethings about the more general setting where the triangle inequality does not hold. Can we do something with predictions in this case?

**Limitations:**

yes

**Strengths And Weaknesses:**

Strengths
- This is the first learning-augmented algorithm for TSP with realistic predictions. The algorithm can in particular be used to robustify NN-based predictors.
- The paper also discusses running times and practicability of their results, which I think is very important for a problem such as TSP.
- The paper is both interesting for the learning-augmented community and for the neural combinatorial optimization community, which are both active areas of research and well-established in ICML.
- The paper is well-written and easy to follow.

Weaknesses
- The only weakness that I see is that the technique is somewhat close to the previous work on approximation algorithms for subset selection problems by Antoniadis et al. However, as the authors point out, TSP is a different problem and has its own unique challenges.

---

> ### Author Rebuttal · Authors · 2026-03-31
>
> Thank you for your review and your feedback.
>
> To answer your question, general TSP is known to be inapproximable without the triangle inequality, i.e., no polynomial-time algorithm can achieve a finite approximation ratio for non-metric TSP, unless P=NP. Unfortunately, by a simple adaptation of this classical inapproximability argument, one can show that no polynomial-time algorithm can achieve a finite approximation ratio even if provided with predictions with arbitrarily small prediction error. We will sketch this argument in the next revision of our paper.

---

> > ### Author Rebuttal · Reviewer_PtpL · 2026-03-31
> >
> > I thank the authors for their rebuttal and for answering my question. I will continue to support this paper.

---

### Official Review · Reviewer_9CrJ · 2026-02-24

**Soundness:** 3
**Presentation:** 3
**Significance:** 3
**Originality:** 3
**Overall Recommendation:** 4
**Confidence:** 4

**Summary:**

This paper studies learning-augmented algorithms for metric TSP, where the input includes an edge heatmap prediction and the goal is to convert it into a feasible tour with performance that degrades gracefully with prediction error. The authors define a weighted $L_1$ error and give a Christofides-style decoding that achieves $\mathbb{E}[w(X)] \le w(X^*) + 2\eta$ in $O(n^3)$ time.   They further propose faster variants with a larger constant on $\eta$ and empirically show their decoding (notably a CHR+ variant) is the only evaluated "solution search" method that consistently leverages heatmaps to outperform vanilla Christofides across predictors and datasets.

**Compliance With Llm Reviewing Policy:**

Affirmed.

**Final Justification:**

My concerns have been addressed. The theoretical part is likely to benefit the largely application-driven ML4TSP field. That said, I still strongly encourage the authors to further strengthen the coverage and discussion of related work, even if a lighter experimental scope is acceptable for theory-driven papers. I therefore maintain my original score.

**Key Questions For Authors:**

1. Since CHR$^+$ is selected as the primary decoder empirically, is there an analogous prediction-error notion and proof technique that yields a bound for Christofides run on modified weights $w'(e)=w(e)(1-q(e))$ (or related continuous reweighting)?
2. The analysis hinges on $\eta(p,X^\*)$ defined w.r.t. an (unknown) optimal tour $X^*$. In practical deployments where OPT is unknown and the heatmap may be miscalibrated, how sensitive is the decoding to systematic biases vs. edge-ranking accuracy, and is it possible to relate performance to an observable surrogate (e.g., error w.r.t. a near-optimal tour or marginal constraints)?
3. The empirical section would benefit from a presentation format more aligned with standard practice in the NCO literature. In particular, it would be helpful to include a consolidated results table (similar in spirit to the current Table 1), reporting **both absolute (obj.) and relative (gap) performance** on **widely used benchmarks** (e.g., the well-known open-source benchmarks with 1280 instances for TSP-50/100 and 128 instances for TSP-500/1k under uniform distributions adopted by e.g., Joshi et al., 2022; GNN-GLS; AM; POMO; DIMES; T2T; Fast-T2T; COExpander; Sym-NCO; ML4CO-Bench-101; GLOP) alongside more **commonly compared methods**. This would improve comparability and make it easier to contextualize the proposed approach within existing work. A standardized, side-by-side reporting format **placed in the main body of the paper** would clarify where gains stem from modeling, decoding, or evaluation protocol, and make the empirical claims more transparent.

| Method                      | Decoder     | Obj. (Absolute Tour length) | Optimality Gap (%) | Per-instance Time (s) |
| --------------------------- | ----------- | --------------------------- | ------------------ | --------------------- |
| // Exact solver (e.g. Concorde) to position the optimality                             |             |                             |                    |                       |
|...|||||
| // non-heatmap baseline (e.g. one or two RL-driven methods as comparative baselines)    |             ||                    |                       |
|...|||||
| //heatmap-decoding paradigm (core results with clear ablative decoding settings) |             |                             |                    |                       |
| GCN4CO                      | Greedy      |                             |                    |                       |
| GCN4CO                      | MCTS        |                             |                    |                       |
| GCN4CO                      | Beam        |                             |                    |                       |
| GCN4CO                      | Greedy+2Opt |                             |                    |                       |
| GCN4CO                      | CHR+     |                             |                    |                       |
| DIFUSCO                     |             |                             |                    |                       |
| ...                         |             |                             |                    |                       |
| GNNGLS                      |             |                             |                    |                       |
| ...                         |             |                             |                    |                       |
| DIMES                       |             |                             |                    |                       |
| ...                         |             |                             |                    |                       |
| Fast-T2T                    |             |                             |                    |                       |
| ...                         |             |                             |                    |                       |
| UTSP                        |             |                             |                    |                       |
| ...                         |             |                             |                    |                       |
| SoftDist                    |             |                             |                    |                       |
| ...                         |             |                             |                    |                       |


4. Since both the theoretical analysis and empirical evaluation are restricted to 2D Euclidean (metric, symmetric) TSP instances, it is unclear how well the proposed framework generalizes beyond this setting. In particular, would the guarantees or practical effectiveness persist under asymmetric or non-metric cost structures? Moreover, many real-world routing problems involve additional constraints (e.g., capacity, time windows) as in various VRP formulations. It would be valuable for the authors to discuss whether the learning-augmented decoding paradigm can be practically extended to such settings. Even a conceptual sketch or outlook in this regard would be beneficial.


**Note:** I appreciate the paper's main contribution, apparently the theoretical derivations and what appears to be an early principled TSP decoding approach with prediction-sensitive guarantees. Accordingly, I assign a positive score at this stage. However, **maintaining an accept recommendation is contingent on the authors' responses to the concerns, suggestions, and questions raised above, and their commitment to a major revision of the manuscript and experimental efforts.**

**Limitations:**

There is currently no explicit discussion of limitations. A separate section outlining the limitations of the present work would be necessary.

**Strengths And Weaknesses:**

### Strengths

1. **Good general presentation.** The paper is well written, with clearly structured mathematical formulations and coherent textual and visual presentations that make the technical contributions rigorous while easy to follow.

2. **New (if not the only) prediction-sensitive guarantees for TSP decoding.** The main bound $\mathbb{E}[w(X)] \le w(X^*) + 2\eta$ directly links tour quality to a weighted $L_1$ distance between the heatmap and an (optimal) tour, addressing the lack of such explicit end-to-end guarantees in prior heatmap-to-tour pipelines.


3. **Principled algorithmic integration with classical Christofides.** The decoding biases the MST toward predicted edges and then performs an odd-vertex join (perfect matching) using original weights, with analysis tracking false positives/negatives through the MST and join costs.

5. **Practical relevance to neural TSP solving.** The paper explicitly targets the neural combinatorial optimization (NCO) pipeline's bottleneck: turning edge probabilities into tours. It positions itself between weak greedy/beam decoders and heavy MCTS, enriching the toolbox for existing NCO decoders.

### Weaknesses with Advice

1. The main theorem is stated for $\mathbb{E}[w(X)]$ and the heatmap-to-set conversion samples edges independently according to $p$, which can be a mismatch for structured predictors (e.g., neural heatmaps with strong correlations).
2. The experiments adopt CHR+ as the primary decoding strategy because it performs best, but the theoretical analysis is developed for the subset-to-zero-weight mechanism (Alg1/Alg1Top), leaving the strongest empirical variant without matching guarantees.
3. The related work section is largely incomplete, even if considering merely within the neural TSP solving literature, where many prior approaches also adopt heatmap-decoding paradigms. The authors should substantially expand this discussion (e.g. a separate section in the appendix) by incorporating more **recent and advanced** NCO methods (at minimum those focused specifically on neural TSP) and positioning their contribution relative to them. In addition, stronger empirical evidence would come from plug-and-play evaluations that substitute the proposed decoding scheme into existing pipelines (e.g., Fast-T2T + greedy vs. Fast-T2T + CHR$^+$), enabling a clearer attribution of performance gains to the decoding strategy itself.


- POMO: Policy Optimization with Multiple Optima for Reinforcement Learning, NeurIPS 2020
- Generalize a Small Pre-trained Model to Arbitrarily Large TSP Instances, AAAI 2021
- MatNet: Matrix Encoding Networks for Neural Combinatorial Optimization, NeurIPS 2021
- Learning Collaborative Policies to Solve NP-hard Routing Problems, NeurIPS 2021
- Graph Neural Network Guided Local Search for the Travelling Salesperson Problem, ICLR 2022
- Sym-NCO: Leveraging Symmetricity for Neural Combinatorial Optimization, NeurIPS 2022
- DIMES: A Differentiable Meta Solver for Combinatorial Optimization Problems, NeurIPS 2022
- T2T: From distribution learning in training to gradient search in testing for combinatorial optimization, NeurIPS 2023
- BQ-NCO: Bisimulation quotienting for efficient neural combinatorial optimization, NeurIPS 2023
- Unsupervised learning for combinatorial optimization needs meta learning, ICLR 2023
- Fast T2T: Optimization Consistency Speeds Up Diffusion-Based Training-to-Testing Solving for Combinatorial Optimization, NeurIPS 2024
- MVMoE: Multi-Task Vehicle Routing Solver with Mixture-of-Experts, ICML 2024
- A diffusion model framework for unsupervised neural combinatorial optimization, ICML 2024
- GLOP: Learning Global Partition and Local Construction for Solving Large-scale Routing Problems in Real-time, AAAI 2024
- COExpander: Adaptive Solution Expansion for Combinatorial Optimization, ICML 2025
- GOAL: A Generalist Combinatorial Optimization Agent Learner, ICLR 2025
- Preference optimization for combinatorial optimization problems, ICML 2025
- UniCO: On Unified Combinatorial Optimization via Problem Reduction to Matrix-Encoded General TSP, ICLR 2025
- BOPO: Neural Combinatorial Optimization via Best-anchored and Objective-guided Preference Optimization, ICML 2025
- DualOpt: A Dual Divide-and-Optimize Algorithm for the Large-scale Traveling Salesman Problem, AAAI 2025
- UnifyML4TSP: Drawing Methodological Principles for TSP and Beyond from Streamlined Design Space of Learning and Search, ICLR 2025
- ML4CO-Bench-101: Benchmark Machine Learning for Classic Combinatorial Problems on Graphs, NeurIPS 2025
- UCPO: A Universal Constrained Combinatorial Optimization Method via Preference Optimization, AAAI 2026

**Note** this is only a rough list for your quick reference. For a 2026 submission in the NCO domain, these references (if not more) are among the most fundamental works that are indispensable for building an up-to-date literature retrospective and demonstrating authoritative expertise and proficiency within this community.

5. **Scope limitations of the experimental evaluation and broader applicability.** Beyond the limited ablation comparison mentioned in point 4 above, the results focus on Euclidean instances (synthetic and TSPLIB) and specific predictors/configurations, and the paper notes discrepancies with original DIFUSCO results likely due to disabled inference-time settings (e.g., sparsification), which complicates apples-to-apples comparison and may affect external validity. The applicability of the proposed theoretical contribution towards more CO problems are yet to be verified, e.g., many node-oriented tasks (Max independent set, Max vertex cover, Max cut, etc.) could also be learned via supervised heatmap prediction, which then can be consumed by decoding strategies.
6. Minor point: the naming "GNN-AR" for the work of Joshi et al. (2022) is potentially misleading. In the neural combinatorial optimization literature, "AR" conventionally refers to autoregressive solvers (e.g., AM, POMO, Sym-NCO, MatNet) that construct tours sequentially and are typically trained with RL, rather than to a decoding strategy. Using this label may therefore cause confusion within the community. Recent benchmarking studies instead refer to this method as direct as "GCN4CO" or "GNN4TSP", which would be more appropriate as aligning with established convention.
7. The current title is somewhat uninformative, particularly within the mainstream NCO literature where application-driven studies dominate. It may be beneficial to adopt a more precise, technically grounded title that clearly positions the contribution as a **theoretically guaranteed, while generally applicable decoding strategy, one that is largely orthogonal to, and can be paired with, a wide range of existing heatmap-based predictors, achieving most time performance gains**. This would better communicate both the methodological novelty and its intended role in the neural TSP pipeline.

---

> ### Author Rebuttal · Authors · 2026-03-31
>
> ### Regarding Weakness 1:
> We are not sure what the reviewer exactly means by a heatmap with strong correlations---a heatmap in our paper is just a sequence of numbers, not random variables. However, just to clarify a potential confusion: the expectation in our guarantee is over the internal (test-time) randomness of our decoding algorithm, and *not* over any test-time randomness of a predictor nor over a random choice of training samples. Therefore, one can easily apply standard probability amplification to turn our in-expectation guarantees to with-high-probability guarantees.
>
> ### Regarding Weakness 2 and Question 1:
> The guarantees of Theorem 1.1 apply also to CHR+. In fact, our analysis translates easily by replacing the characteristic vector of the prediction $P$ with the heatmap, e.g., instead of $w({X}\textbackslash P)$ we would write $\sum_{e\in X} w(e)(1-p(e))$. We have chosen to present our analysis with $P$ sampled according to the heatmap due to its clarity. We thank the reviewer for pointing this out---we will add a note that the analysis translates to CHR+ as well.
>
> ### Regarding Weakness 3:
> We agree with the reviewer that the related work section is currently limited, and we will substantially expand it in the final version to include a more comprehensive overview of neural combinatorial optimization approaches, particularly recent advances in neural TSP solvers, starting from the rich set of references highlighted by the reviewer. That said, we would like to clarify the scope of our contribution. The primary goal of this work is not to propose a new state-of-the-art end-to-end neural TSP solver, but rather to study and improve the decoding stage given a predicted heatmap. Many existing approaches integrate additional components beyond heatmap prediction, which makes direct comparisons at the pipeline level less informative, as performance differences would conflate the quality of the whole pipeline with that of the decoder.
>
> Finally, we emphasize that our paper is primarily theory-driven, and the experimental section is intended as a proof of concept rather than a comprehensive empirical benchmark. For instance, we decided to not include FastT2T because it builds on top of a diffusion model, like DIFUSCO, which we already included.
>
> ### Regarding Weakness 5:
> We thank the reviewer for raising these important points. To broaden the scope of the experimental evaluation, we have included additional experiments on non-Euclidean (general metric) TSP instances---please see our response to review sp3v.
>
> We remark that for other CO problems, such as Independent Set or Vertex Cover, there is already existing literature, in particular Antoniadis et al. ICLR'25. Our framework is specifically designed for TSP, which is a more challenging problem than the two aforementioned ones, because in order to use ideas from classical constant-factor approximation algorithms we need to ensure triangle inequality, which gets destroyed once we adjust edge weights according to predictions.
>
> ### Regarding Weaknesses 6 and 7:
> We thank the reviewer for these comment. We will change the naming of the Joshi et al's predictor to adhere with the literature, and we will change the title to a more informative one: ``TSP with predictions: heatmap to tour with provable guarantees''
>
> ### Regarding Question 2:
> See lines 58--65 in the second column: ``The statement of Theorem 1.1 holds for all TSP tours X* simultaneously (optimal or not), which can be interpreted as follows: As far as p has a small error with respect to some TSP tour of a small weight, the algorithm will also find a tour of a small weight.''
>
> ### Regarding Question 3:
> We will revise all tables as suggested by the reviewer. Regarding the dataset, we have replaced our previous instances with the standard Euclidean datasets used in GNNGLS and GNN4CO to ensure a consistent evaluation. All figures and tables will be updated in the final version. The experiments with updated datasets are still running, but the majority of the revised results are already available in our repository https://github.com/anonym-doubleblind/ICML26, demonstrating that the conclusions of the paper remain unchanged.
>
> ### Regarding Question 4:
> We refer to our responses to reviewers PtpL and sp3v for discussion and additional experimental results extending our results to other variants of TSP. Our approach might be useful also in VRP, especially when approached using Route first--Cluster second strategies (see https://doi.org/10.1016/0305-0483(83)90033-6). However, extension of our theoretical results to VRP is not immediate and would require significant research effort.

---

> > ### Author Rebuttal · Reviewer_9CrJ · 2026-04-01
> >
> > I thank the authors for their rebuttal. I understand this work is mainly theory-driven, so a substantial expansion of discussions on related neural TSP solvers would be satisfactory. I maintain my positive recommendation of this paper.

---

### Official Review · Reviewer_sp3v · 2026-03-05

**Soundness:** 3
**Presentation:** 3
**Significance:** 3
**Originality:** 3
**Overall Recommendation:** 4
**Confidence:** 3

**Summary:**

This paper presents the first learning-augmented approximation algorithms for the TSP, addressing a critical gap in the intersection of machine learning and combinatorial. The core contribution is a set of algorithms that transform the heatmaps into feasible TSP tours with provable theoretical guarantees. Empirically, the authors validate the proposed algorithm against strandart baselines and demonstrate the effectiveness of the proposed method.

**Compliance With Llm Reviewing Policy:**

Affirmed.

**Final Justification:**

I have carefully reviewed the authors’ rebutttals, and my concerns have been adequately addressed.

**Key Questions For Authors:**

1. For the near-linear time metric TSP algorithm (Theorem 1.2), is there a way to reduce the multiplicative factor on eta by using a better approximation for perfect matching or a more sophisticated MST biasing strategy?
2. The paper proposes CHR+ as a drop-in replacement for the solution search stage in neural combinatorial optimization pipelines. Have you integrated CHR+ into an end-to-end ML + optimization pipeline (i.e., training a GNN to predict heatmaps and using
CHR+ as the solution search step), and if so, what is the end-to-end performance compared to state-of-the-art neural TSP solvers on large instances? If not, what are the main challenges to such an integration?

**Limitations:**

There is no discussion of limitations.

**Strengths And Weaknesses:**

Strengths:
1. This is the first work to develop learning-augmented approximation algorithms for TSP with explicit, quantitative guarantees on prediction error, establishing a foundational result for learning-augmented TSP research.
2. The work bridges neural combinatorial optimization and classical TSP approximation by fixing a key flaw in existing pipelines: the disconnect between ML heatmap predictions and the solution search stage. Unlike MCTS or greedy search, the proposed algorithms are lightweight and heavily leverage ML predictions while retaining Christofides’ worst-case performance.

Weaknesses:
1. The empirical evaluation focuses on Euclidean TSP only, with predictors limited to GNNs and simple heuristics.
2. The core algorithm relies on an exact minimum-weight perfect matching step, which is computationally expensive for very large instances.

---

> ### Author Rebuttal · Authors · 2026-03-31
>
> We thank the reviewer for their feedback and questions. We provide the response to each of them.
>
> ### Regarding Weakness 1:
> To our best knowledge, different variants of GNNs are state-of-the art approach for generating TSP heatmaps---we do not understand why using them is considered a weakness. Regarding focus on Euclidean instances, this is due to the fact that the predictors we tested were trained on such instances, and do not generalize particularly well to non-Euclidean settings. However, in the last few days we run a small non-Euclidean experiment: We selected TSPLIB instances with less then 1000 nodes that satisfy the triangle inequality (up to $10^{-6}$ tolerance). This results in a (small) library of 15 instances. We generated predictions for these instances using two approaches: the non-neural heuristic SoftDist and the neural network GNNGLS. The latter does not explicitly require its inputs to be Euclidean, but it has been trained only on Euclidean instances and it does not generalize particularly well. The other two neural predictors tested in our paper do not work in the non-Euclidean setting (because they use node coordinates as features), so we did not include them in this experiment. Results are available in this table: https://github.com/anonym-doubleblind/ICML26/blob/main/table_new.png. Even though the performance gains are less pronounced than in the Euclidean setting, our CHR$^+$ remains the most robust decoder across all instances.
>
> ### Regarding Weakness 2:
> Indeed, in our proof-of-concept experiments we use a relatively slow Python implementation of the Edmonds min-weight perfect matching algorithm. However, there exist much faster implementations, e.g., Blossom V (https://doi.org/10.1007/s12532-009-0002-8) that runs within few seconds on graphs with hundreds of thousands of vertices. Moreover, for even larger graphs, we propose algorithms that use approximate matching algorithms and run in near-linear time, while achieving the same qualitative guarantees (up to a constant factor in the dependence on prediction error).
>
> ### Regarding Question 1:
> The constant 11 can be improved to $6+\epsilon$ by using a $(1+\epsilon)$-approximation algorithm for matching (which exists, e.g., for the Euclidean setting), through a simple change in the analysis in line 767.
>
> It is not clear if an improvement beyond 6 is possible. It might require a different analysis (e.g., comparison to a different low-cost T-join that is more aligned with the prediction), or changes to the algorithm itself (e.g., a different MST biasing strategy as suggested by the reviewer).
>
> However, while getting a better constant here is surely an interesting theoretical question, we prefer to think of this theorem as a *qualitative* description of performance. For instance, in Figure 5, the empirical cost of Algorithm 1 anyway looks more like $OPT + 0.5\eta$, instead of $OPT + 2\eta$ which is guaranteed by Theorem 1.1 and shown to be tight in Section D.2.
>
> ### Regarding Question 2:
> As a direction for future work, it would be natural to design and train end-to-end architectures with CHR$^+$ built in.  We have not explored this direction yet, because the primary goal of this paper is to introduce CHR$^+$ and demonstrate its effectiveness, and it is *not* to develop a new end-to-end pipeline. Accordingly, our experimental evaluation is intended as a proof of concept, rather than as a fully optimized, state-of-the-art end-to-end neural TSP solver. If one wish to train an end-to-end solver, e.g., GNN + CHR$^+$, one has to deal with the non-differentiability of CHR$^+$. It might be non trivial, but there are approaches dealing with similar issues, e.g.,
> https://openreview.net/forum?id=BkevoJSYPB

---

> > ### Author Rebuttal · Reviewer_sp3v · 2026-04-01
> >
> > Thank you for your comments. At this stage, I have no further concerns and will keep my positive recommendation.

---

### Decision · Program_Chairs · 2026-04-30

**Decision:**

Accept (regular)

**Comment:**

This work considers the Traveling Salesperson Problem (TSP) and proposes learning-augmented approximation algorithms with theoretical guarantees. Reviewers all stressed the quality of the presentation, the impact of the results (bridging neural combinatorial optimization and learning-augmented communities) as well as the neat theoretical guarantees on the prediction error. Concerns about the practical applications and computational cost were adequately addressed during the rebuttal phase and of minor importance due to the theoretical nature of the contribution. This motivates acceptance.